# Evolving Minds: Logic-Informed Inference from Temporal Action Patterns

**Chao Yang** [1]  **Shuting Cui** [1]  **Yang Yang** [1]  **Shuang Li** [1]

## Abstract

Understanding human mental states—such as intentions and desires—is crucial for natural AI-human collaboration. However, this is challenging because human actions occur irregularly over time, and the underlying mental states that drive these actions are unobserved. To tackle this, we propose a novel framework that combines a logic-informed temporal point process (TPP) with amortized variational Expectation-Maximization (EM). Our key innovation is integrating logic rules as priors to guide the TPP's intensity function, allowing the model to capture the interplay between actions and mental events while reducing dependence on large datasets. To handle the intractability of mental state inference, we introduce a discrete-time renewal process to approximate the posterior. By jointly optimizing model parameters, logic rules, and inference networks, our approach infers entire mental event sequences and adaptively predicts future actions. Experiments on both synthetic and real-world datasets show that our method outperforms existing approaches in accurately inferring mental states and predicting actions, demonstrating its effectiveness in modeling human cognitive processes.

## 1. Introduction

Rapid advancement of AI has spurred interest in developing autonomous agents that collaborate with humans in tasks such as healthcare, education, and robotics (Carroll et al., 2019; Puig et al., 2020; Strouse et al., 2021). Effective collaboration requires AI agents to understand human actions and infer latent intentions—mental states that drive behavior but remain unobserved. This capability is essential for enabling agents to provide timely, context-aware assistance in dynamic environments.

Human behavior, though complex, often follows simple, generalizable logic. In Fig.1, preparing oatmeal involves sequential steps (e.g., taking a cup, adding oats, and pouring milk) governed by underlying intentions like wanting oatmeal soaked in milk (Damen et al., 2018). Logic rules that define the mutual relationship between intentions and actions provide compact, interpretable representations of this knowledge. Using such rules, AI systems can better predict behavior and infer mental states.

Prior work has focused on forecasting actions from observed sequences (Abu Farha et al., 2018; Cramer et al., 2021; Darvish et al., 2020), but critical challenges remain. First, actions and mental states occur irregularly in time: intentions shift unexpectedly (e.g., abandoning a task due to distraction), and the time intervals between actions encode contextual cues about these shifts. Ignoring temporal irregularity risks misinterpreting intentions, as delays or abrupt transitions often signal evolving goals (Hu & Clune, 2023; Zolotas & Demiris, 2022). Second, mental states are dynamic and interdependent: past actions influence intention formation, while intentions guide future behavior. Inferring the entire sequence of mental events—when and what intentions occur—requires modeling these mutual dependencies with precise temporal granularity. Traditional methods struggle to disentangle irregular, intertwined processes, leading to oversimplified or stochastic predictions.

To address these challenges, we propose a novel framework that unifies logic-informed temporal point processes (TPPs) and amortized variational inference to model the bidirectional dynamics between irregular human actions and latent mental states. Our approach jointly learns to infer intentions, predict future actions, and discover interpretable logic rules—all within a single, cohesive learning paradigm.

Central to our framework is a logic-informed TPP, where domain knowledge is encoded as probabilistic logic rules that shape the intensity functions governing actions and mental events. These rules (Li et al., 2020; Mei et al., 2020), act as priors to constrain the learning space, reducing reliance on vast datasets while preserving interpretability. Crucially, our model captures two-way interactions: past actions probabilistically trigger or reset intentions (e.g., prolonged inactivity may signal a shift in goals), while inferred intentions guide the likelihood and timing of future actions.

---

[1]School of Data Science, The Chinese University of Hong Kong (Shenzhen), Shenzhen, China. Correspondence to: Shuang Li <lishuang@cuhk.edu.cn>.

*Proceedings of the $42^{nd}$ International Conference on Machine Learning*, Vancouver, Canada. PMLR 267, 2025. Copyright 2025 by the author(s).

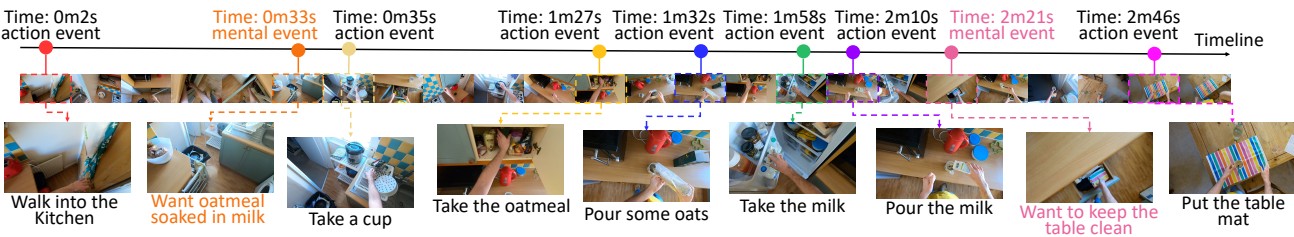

*Figure 1.* An illustrative example of complex interaction between intentions and action events: at 0m33s, the individual forms the intention to have oatmeal with milk, and at 2m21s, another intention arises, focused on keeping the table clean. These intentions are driven by past actions and consequently result in a sequence of future actions.

To address the challenge of latent mental states, we develop an amortized variational EM algorithm that alternates between the following steps:

**E-step:** Refining an amortized inference network to approximate the posterior distribution of mental states. The inference network is modeled as a discrete-time renewal process (DT-RP) with neural hazard functions, effectively capturing intention resetting and long-term dependencies.

**M-step:** Optimizing continuous TPP parameters and discrete logic rules. The logic rule set evolves dynamically during training: predefined rules bootstrap learning, while an automated column generation strategy (Barnhart et al., 1998; Li et al., 2021) expands the rule space with data-driven patterns (e.g., discovering new intention-action correlations).

The integration of symbolic priors with neural inference allows our model to adapt to individual behavioral nuances while maintaining explainability. Once trained, the model predicts future actions by inferring mental events and sampling action timing/type. A backtracking mechanism is used to ensure temporal consistency by checking for mental state shifts between observed and predicted actions, aligning predictions with inferred intentions.

Our contributions can be summarized as follows: (*i*) We propose a logic-informed TPP framework that captures irregularly spaced actions and latent mental states. By integrating logic rules as symbolic priors, our approach enhances data efficiency and ensures model interpretability. (*ii*) We introduce a variational EM algorithm with a DT-RP for efficient and scalable inference of latent mental event sequences. The M-step jointly optimizes model parameters and the logic rule set, leveraging automated rule discovery to reveal novel and overlooked intention-action patterns. (*iii*) Extensive experiments demonstrate the effectiveness of our method in inferring complex mental event sequences and accurately predicting future actions, advancing human-centric AI.

## 2. Preliminaries

We present key foundations for modeling event sequences. TPPs will be used to model the intertwined generative pro-

cesses of action and mental events, while DT-RPs will be introduced to approximate latent mental state posteriors.

### 2.1. Temporal Point Processes (TPPs)

**Intensity Function**   TPPs model the occurrence of events in continuous time through intensity functions that encode dependencies on historical context. Let $\mathcal{X}$ denote event types (e.g., actions like *adding oats* or mental states like *cooking intent*). Each event $e_i := (t_i, x_i)$ occurs at time $t_i$ with type $x_i \in \mathcal{X}$, and the history up to time $t$ is $\mathcal{H}(t) = \{e_j \mid t_j < t\}$. The conditional intensity $\lambda_x(t \mid \mathcal{H}(t))$ quantifies the instantaneous rate of event $x$ at time $t$:

$$\lambda_x(t \mid \mathcal{H}(t)) = \lim_{\Delta t \to 0} \frac{\mathbb{P}(x \text{ occurs in } [t, t + \Delta t) \mid \mathcal{H}(t))}{\Delta t}.$$

A classic example is the **Hawkes process** (Hawkes, 1971; Zuo et al., 2020), which models self- and mutual-excitation between event types. Its intensity for type $x$ is:

$$\lambda_x(t \mid \mathcal{H}(t)) = \mu_x + \sum_{e_j \in \mathcal{H}(t)} \alpha_{x,x_j} e^{-\beta_{x,x_j}(t-t_j)},$$

where $\mu_x$ is the base rate, $\alpha_{x,x_j}$ defines how strongly a past event $e_j$ (of type $x_j$) excites future events of type $x$, and $\beta_{x,x_j}$ controls the decay of this influence.

**Learning and Sampling**   For observed events $\boldsymbol{e} = \{e_1, \ldots, e_N\}$ over $[0, T]$, the log-likelihood is:

$$\log p_\theta(\boldsymbol{e}) = \sum_{i=1}^{N} \log \lambda_{x_i}(t_i \mid \mathcal{H}(t_i)) - \int_0^T \sum_{x \in \mathcal{X}} \lambda_x(\tau \mid \mathcal{H}(\tau)) d\tau,$$

(1)

where $\theta$ includes TPP parameters (e.g., $\mu_x, \alpha_{x,x_j}, \beta_{x,x_j}$ for Hawkes process). By maximizing this likelihood, we learn the generative dynamics of the events encoded in $\theta$. Sampling events in TPPs often requires iterative methods like thinning algorithm (Ogata, 1981; Rasmussen, 2018), where candidate event times are proposed and accepted/rejected based on the intensity function.

**Modeling Actions and Mental States**   While TPPs like neural Hawkes processes capture complex triggering patterns, their dense parameterization poses challenges. The

pairwise parameters are opaque, making it hard to explain how actions or mental states influence each other. Learning all pairwise interactions requires large datasets, which are scarce in behavioral domains. Moreover, these models cannot easily represent conditional triggering logic (e.g., "if A and B occur, trigger C"), which is essential for capturing complex mental state-behavior relationships. To address these limitations, we integrate **logic rules** as inductive biases, grounding the model in domain knowledge. For example, a rule like "*cook meal ← hunger ∧ available ingredients*" explicitly links desire to actions, pruning irrelevant connections (e.g., setting $\alpha_{x,x_j} = 0$ for irrelevant pairs) and reducing need for dense parameter learning. Crucially, these rules can be predefined or learned in our framework, adapting to new scenarios and improving generalization. For more details, see Sec. 3.3.

## 2.2. Discrete-Time Renewal Processes (DT-RPs)

In our method, we propose using DT-RPs to approximate the intractable posterior distribution of latent mental processes, for ease of implementation.

**Discretization and Hazard Rates**   Time horizon $T$ is divided into $K = \lceil T/\Delta t \rceil$ grids $V_k = ((k-1)\Delta t, k\Delta t]$. For each event type $x$ (e.g., mental states), the hazard function $h_x(k) \in [0,1]$ indicates the probability of $x$ occurring in $V_k$, given no $x$-events since last occurrence at grid $k_0$, i.e.,

$$h_x(k) = \mathbb{P}\left(x \text{ in } V_k \mid \text{No } x \text{ in } V_{k_0+1}, \ldots, V_{k-1}\right). \quad (2)$$

Recall that in TPPs, the hazard rate corresponds to the instantaneous intensity function, which can exceed 1 (Rasmussen, 2018). In contrast, in DT-RPs, the hazard represents the conditional probability of an event occurring within a discrete time interval and thus restricted to values between 0 and 1. Correspondingly, the survival probability $S_x(k)$, indicating likelihood of no $x$-events from $k_0 + 1$ to $k$, is computed as

$$S_x(k) = \prod_{\tau=k_0+1}^{k} (1 - h_x(\tau)). \quad (3)$$

From the definition, each event $x$ "restarts its own clock" upon occurrence, resetting the survival function to 1 by updating $k_0$.

**Learning and Sampling**   Given the hazard function $h_x(x)$, we sample the next $x$-event according to Alg. 2 in Appendix. A.1, using inverse transform sampling method. With a hazard model parametrized by $\phi$, the likelihood for binary event indicators $\boldsymbol{o}(k) \in \{0,1\}^{|\mathcal{X}|}$ over $K$ grids is

$$q_\phi(\boldsymbol{o}) = \prod_{k=1}^{K} \prod_{x \in \mathcal{X}} h_x(k)^{o_x(k)} (1 - h_x(k))^{1-o_x(k)}, \quad (4)$$

where $o_x(k) = 1$ indicates that an event $x$ occurs in the $k$-th interval and $o_x(k) = 0$ otherwise. This grid-wise factorization enables efficient training by decomposing likelihoods across time and event types.

**Efficient Computation and Sampling**   We use DT-RPs to approximate mental state posteriors because they allow for efficient sampling and likelihood evaluation. Unlike TPPs, which rely on costly iterative thinning and numerical integration, DT-RPs divide time into intervals. This approach enables fast sampling through cumulative hazards and calculates event probabilities separately for each interval. By prioritizing simplicity and speed over continuous-time precision, DT-RPs align well with the amortized variational EM framework's need for rapid latent event sampling and repeated likelihood evaluations.

## 3. Amortized Variational EM

Human decision making is inherently shaped by complex cognitive processes that remain largely unobservable. We propose an amortized variational EM framework built upon event data models. This framework, visualized in Fig. 2, can learn coupled generative processes of actions and mental events, infer latent mental events and predict future actions. Consider the following action and mental events:

- **Action:** Observed sequence $\boldsymbol{a} = \{(t_i^a, x_i^a)\}_{i=1}^{N_a}$, where $x_i^a \in \mathcal{A}$ (e.g., "*boil water*").

- **Mental:** Latent sequence $\boldsymbol{m} = \{(t_i^m, x_i^m)\}_{i=1}^{N_m}$, where $x_i^m \in \mathcal{M}$ (e.g., "*cooking intent*").

We use the logic-informed TPPs (TLPP) to model the interleaved generative processes between action and mental events, parameterized by $p_\theta(\boldsymbol{a}, \boldsymbol{m})$, where $\theta$ contains both continuous logic model parameters and the discrete rule set. Details of the logic-informed model is provided in Sec. 3.1. To approximate the intractable posterior for latent mental states $p_\theta(\boldsymbol{m} \mid \boldsymbol{a})$, we use a DT-RP as the variational distribution $q_\phi(\boldsymbol{m} \mid \boldsymbol{a})$. The parameters $\phi$ are amortized across sequences via a neural encoder:

$$q_\phi(\boldsymbol{m} \mid \boldsymbol{a}) = \text{NN}_\phi(\boldsymbol{a}), \quad (5)$$

where $\text{NN}_\phi$ employs an attention-mechanism to extract embeddings from $\boldsymbol{a}$, dynamically shaping DT-RP's hazard. Details of this neural hazard model are provided in Sec. 3.2.

Our goal is to optimize both $q_\phi(\boldsymbol{m} \mid \boldsymbol{a})$ that enables latent mental event inference and $p_\theta(\boldsymbol{a}, \boldsymbol{m})$ that enables rule mining and events dynamic learning, through Evidence Lower Bound (ELBO), which serves as the objective for training:

$$\mathcal{L}(\theta, \phi; \boldsymbol{a}) = \mathbb{E}_{q_\phi} \left[\log p_\theta(\boldsymbol{a}, \boldsymbol{m})\right] - \mathbb{E}_{q_\phi} \left[\log q_\phi(\boldsymbol{m} \mid \boldsymbol{a})\right]$$
$$= \mathbb{E}_{q_\phi} \left[\log p_\theta(\boldsymbol{a}, \boldsymbol{m})\right] + \mathbb{H}\left[q_\phi(\boldsymbol{m} \mid \boldsymbol{a})\right]. \quad (6)$$

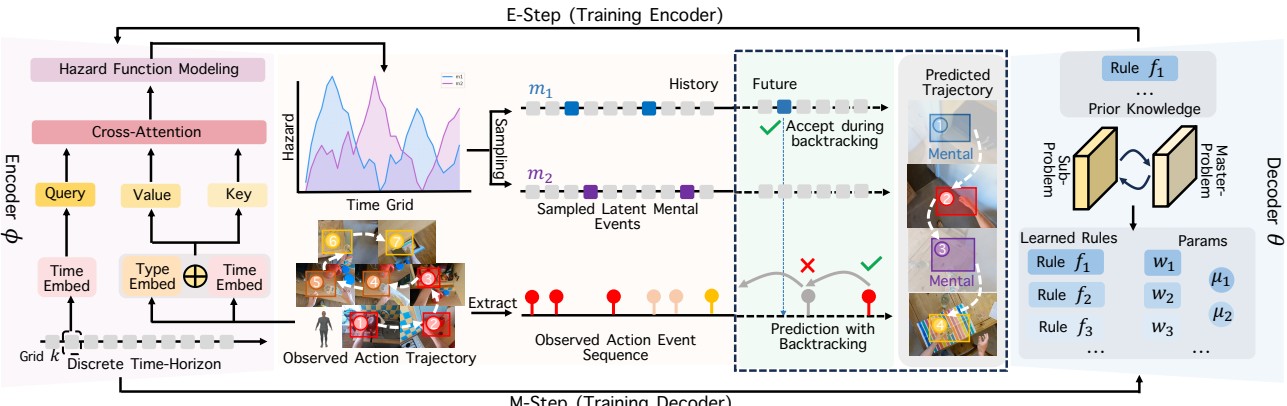

*Figure 2.* Model architecture. **E-Step**: freezing $\theta$ to optimize $\phi$ for training encoder. **M-Step**: freezing $\phi$ to optimize $\theta$ for training decoder. **Prediction**: the prediction with backtracking mechanism is presented in dashed box.

## 3.1. Decoder (Logic-Informed TPP $p_\theta(\boldsymbol{a}, \boldsymbol{m})$)

The decoder models joint dynamics of action and mental events, which integrates domain-specific temporal logic rules to capture the structured dependencies between events.

**Event Representation and Temporal Logic** We unify action and mental event types into a single set $\mathcal{X} = \mathcal{A} \cup \mathcal{M}$, where $x \in \mathcal{X}$ is represented as a Boolean variable:

$$x(t) = \begin{cases} 1 & \text{if an event of type } x \text{ occurs at time } t \\ 0 & \text{otherwise} \end{cases}$$

For observed actions $\boldsymbol{a}$ and inferred mental states $\boldsymbol{m}$, these variables are grounded as $x^a(t_i^a)$ and $x^m(t_i^m)$.

Domain knowledge is encoded as temporal logic rules $\mathcal{F}$, which define structured dependencies between events. Each horn rule (i.e., "if-then" rule) $f \in \mathcal{F}$ follows:

$$f : x_{\text{head}} \leftarrow \underbrace{\left( \bigwedge_{x^u \in \mathcal{X}_f} x^u \right) \bigwedge \left( \bigwedge_{R_j \in f} R_j(x^u, x^v) \right)}_{\text{Body conditions}}$$

where the head of the rule is the target event $x_{\text{head}} \in \mathcal{X}$ influenced by the rule, the body conditions are the logical conjunction of event occurrences $(x^u)$ and temporal relations $(R_j)$. The temporal relations are constraints like $\text{Before}(x^u, x^v)$, $\text{After}(x^u, x^v)$, or $\text{Equal}(x^u, x^v)$, grounded in event timestamps.

**Dynamic Rule Influence with Decay** The influence of past events on current intensities decays over time. For each rule $f$, the feature $\phi_f(\mathcal{H}(t))$ counts valid event combinations in history $\mathcal{H}(t)$ that satisfy the rule's body, weighted by an exponential decay:

$$\phi_f(\mathcal{H}(t)) = \sum_{\text{valid comb.}} e^{-\beta_f(t - \tau_{\text{last}})}$$

where $\tau_{\text{last}}$ is the most recent event time in the combination, and $\beta_f > 0$ (can either be pre-defined or learned) controls how quickly the rule's influence diminishes. The intensity for event type $x$ combines base rates and rule contributions:

$$\lambda_x(t \mid \mathcal{H}(t)) = \mu_x + \sum_{f \in \mathcal{F}_x} w_f \phi_f(\mathcal{H}(t))$$

where $\mu_x$ is the base rate, $w_f$ is the rule weight, and $\mathcal{F}_x$ are rules targeting $x$. The model parameters $\theta = \left[ [\mu_x]_{x \in \mathcal{X}}, \mathcal{F}, [w_f]_{f \in \mathcal{F}} \right]$ include continuous base rates $[\mu_x]_{x \in \mathcal{X}}$ and rule weights $[w_f]_{f \in \mathcal{F}}$, as well as discrete rule set $\mathcal{F} = \cup_{x \in \mathcal{X}} \mathcal{F}_x$ (will also be refined during training).

For observed actions $\boldsymbol{a}$ and inferred mental states $\boldsymbol{m}$, the joint likelihood (according to Eq. (1)) is:

$$\log p_\theta(\boldsymbol{a}, \boldsymbol{m}) = \sum_{i=1}^{N_a} \lambda_{x_i^a}(t_i^a \mid \mathcal{H}(t_i^a)) + \sum_{i=1}^{N_m} \lambda_{x_i^m}(t_i^m \mid \mathcal{H}(t_i^m))$$
$$- \int_0^T \sum_{x \in \mathcal{A} \cup \mathcal{M}} \lambda_x(\tau \mid \mathcal{H}(\tau)) d\tau. \tag{7}$$

## 3.2. Encoder (Amortized Posterior $q_\phi(\boldsymbol{m} \mid \boldsymbol{a})$)

The encoder leverages DT-RPs to approximate posterior of latent mental states, combining sampling efficiency and tractable likelihoods for scalable variational inference. Its novelty lies in its neural hazard architecture, dynamically synthesizing action context, mental history, and temporal dynamics to parameterize hazard functions. Our encoder builds the *Neural Hazard Architecture* by integrating the following critical components:

**Action Context via Cross-Attention** Observed actions $\boldsymbol{a} = \{(t_i^a, x_i^a)\}$ are encoded into timeline-aligned embeddings using attention mechanism. Each action is embedded as $\boldsymbol{x}_i = \boldsymbol{u}_i + \boldsymbol{z}_{\text{abs}}(t_i^a)$, where $\boldsymbol{u}_i$ is a learnable type embedding and $\boldsymbol{z}_{\text{abs}}$ encodes absolute timestamps. Discrete time

grids $V_k = ((k-1)\Delta t, k\Delta t]$ are similarly embedded as $\boldsymbol{l}_k = \boldsymbol{z}_{\text{abs}}(k\Delta t)$. A cross-attention layer aligns grids with historical actions:

$$\boldsymbol{S}_k = \text{Softmax}\left(\boldsymbol{Q}\boldsymbol{K}^\top / \sqrt{D}\right)\boldsymbol{V},$$

$$\text{where} \quad \boldsymbol{Q} = \boldsymbol{L}\boldsymbol{W}^Q; \; \boldsymbol{K}, \boldsymbol{V} = \boldsymbol{X}\boldsymbol{W}^K, \boldsymbol{X}\boldsymbol{W}^V \quad (8)$$

where $\boldsymbol{L} = [\boldsymbol{l}_1, ..., \boldsymbol{l}_k, ...]$ and $\boldsymbol{X} = [\boldsymbol{x}_1, ..., \boldsymbol{x}_i, ...]$ represent stacked timeline queries and stacked action keys/values, respectively. The output $\boldsymbol{S}_k \in \mathbb{R}^D$ captures entire action context relevant to grid $V_k$, ensuring temporal alignment critical for inferring mental states.

**Hazard Function Design** The hazard rate $h_x(k)$ for mental event type $x$ at grid $k$ is constructed as:

$$h_x(k) = \sigma(\; \underbrace{f_x(\boldsymbol{S}_k, \boldsymbol{\eta}_k)}_{\text{type-specific emission}} + \underbrace{g_x(k - k_0^x)}_{\text{type-specific timing}} \;), \quad (9)$$

where $f_x : \mathbb{R}^D \times \mathbb{R}^{D'} \to \mathbb{R}$ is a type-specific final layer for each $x$, $g_x : \mathbb{N} \to \mathbb{R}$ is a type-specific MLP modeling elapsed time $k - k_0^x$ since last $x$-event, $\boldsymbol{\eta}_k \in \mathbb{R}^{D'}$ is mental history state (shared across types), updated autoregressively, and $\sigma(\cdot)$ is sigmoid function, constraining $h_x(k) \in (0, 1)$.

**Autoregressive Updates** After sampling $\boldsymbol{o}(k) \in \{0, 1\}^{|\mathcal{M}|}$ via Alg. 2, the mental history updates as

$$\boldsymbol{\eta}_{k+1} = \text{RNN}\left(\boldsymbol{\eta}_k, \boldsymbol{o}(k)\right)$$

resetting $k_0^x$ to $k$ if $o_x(k) = 1$. This enables dynamic, context-sensitive hazard modeling.

This design of hazard function balances parameter efficiency (shared initial layers across event types) with expressivity (type-specific final layers for emissions), which is critical for modeling heterogeneous mental processes. The hazard parameter $\phi$ is learned via E-step, where the entropy term is explicitly defined using hazard-based likelihood, i.e.,

$$\mathbb{H}\left[q_\phi(\boldsymbol{m} \mid \boldsymbol{a})\right] = -\sum_{k=1}^{K}\sum_{x \in \mathcal{M}}\left[h_x(k)\log h_x(k)\right.$$
$$\left. + (1 - h_x(k))\log(1 - h_x(k))\right]. \quad (10)$$

where $h_x(k)$ is parameterized as in Eq. (9). This follows because each term in $q_\phi(\boldsymbol{o})$ is a Bernoulli distribution with parameter $h_x(k)$ as in Eq. (4). Sampling $\boldsymbol{m} \sim q_\phi(\boldsymbol{m} \mid \boldsymbol{a})$ can be efficiently implemented via Alg. 2, which leverages inverse transform sampling over discrete grids.

### 3.3. Adaptive Rule Learning via Column Generation

To accommodate various applications, especially those lacking prior knowledge datasets, we introduce an adaptive rule learning module. It allows us to implement tailored strategies for different scenarios, including autonomous rule learning and the use of predefined rules. We use column generation algorithm which maximizes the expected complete-data log-likelihood over sampled mental states $\boldsymbol{m} \sim q_\phi(\boldsymbol{m} \mid \boldsymbol{a})$, computed during the M-step. The algorithm alternates between optimizing continuous parameters $\tilde{\theta} = \left[[\mu_x]_{x \in \mathcal{X}}, [w_f]_{f \in \mathcal{F}}\right]$ (*Master Problem*) and refines the temporal logic rule set $\mathcal{F}$ (*Sub-Problem*).

**Master Problem (Continuous Parameter Optimization)** Given samples $\left\{\boldsymbol{m}^{(s)}\right\}_{s=1}^{S} \sim q_{\phi^{(t+1)}}(\boldsymbol{m} \mid \boldsymbol{a})$ and the current rule set $\mathcal{F}_j$ optimize the continuous parameters $\tilde{\theta}$ as

$$\tilde{\theta}_j^* = \arg\max_{\tilde{\theta}} \underbrace{\frac{1}{S}\sum_{s=1}^{S}\log p_\theta\left(\boldsymbol{a}, \boldsymbol{m}^{(s)}\right)}_{\text{Expected log-likelihood}} - \Omega\left(\{w_f\}_{f \in \mathcal{F}_j}\right),$$

$$\text{s.t. } w_f \geq 0 \, (\forall f \in \mathcal{F}_j) \quad (11)$$

where the log-likelihood for a sample $(\boldsymbol{a}, \boldsymbol{m}^{(s)})$ is computed as in Eq. (7), $\Omega(\cdot)$ is a differentiable rule complexity regularization term, such as $\alpha \sum_{f \in \mathcal{F}_j} w_f^2$. This regularization encourages simpler rules while preventing overfitting.

**Sub-Problem (Rule Proposal via Sampled Gradients)** New rules are proposed by identifying those with the most negative expected reduced cost, computed over samples $\left\{\boldsymbol{m}^{(s)}\right\}_{s=1}^{S}$:

$$\Delta\mathcal{F} = \{f \notin \mathcal{F}_j \mid \text{Reduced Cost}(f) < 0\}$$

where the reduced cost for rule $f$ is defined as:

$$\text{ReducedCost}(f) = -\frac{1}{S}\sum_{s=1}^{S}\left.\frac{\partial \log p_\theta}{\partial w_f}\right|_{\boldsymbol{m}^{(s)}} + \left.\frac{\partial \Omega}{\partial w_f}\right|_{\tilde{\theta}_j^*}$$

The gradient $\left.\frac{\partial \log p_\theta}{\partial w_f}\right|_{\boldsymbol{m}^{(s)}}$ is derived from the sampled mental events $\boldsymbol{m}^{(s)}$. Among all candidate rules, the one with the most negative reduced cost (i.e., $\arg\min_f \text{ReducedCost}(f)$) is prioritized for addition to $\mathcal{F}_j$ and obtain $\mathcal{F}_{j+1}$. This ensures that the proposed rules maximally improve the expected log-likelihood while accounting for regularization. The algorithm for the column generation is presented in Alg. 3, Appendix. A.2.

### 3.4. Learning Procedure

Now building blocks are prepared, we employ an amortized variational EM algorithm that alternates between E-step and M-step, with algorithm shown in Alg. 4, Appendix. A.3.

- **E-Step: Amortized Variational Inference** Update the inference network $\phi$ to infer mental event trajectories from actions:

$$\phi^{(t+1)} = \arg\max_\phi \mathbb{E}_{q_\phi(\boldsymbol{m}|\boldsymbol{a})}\left[\log p_{\theta^{(t)}}(\boldsymbol{a}, \boldsymbol{m})\right] + \mathbb{H}\left[q_\phi(\boldsymbol{m}|\boldsymbol{a})\right]$$

The gradient of the expectation term is estimated via the REINFORCE estimator with a baseline $c$ (e.g., empirical average) to reduce variance, $\nabla_\phi \mathbb{E}_{q_\phi} [\log p_\theta(\boldsymbol{a}, \boldsymbol{m})] \approx \mathbb{E}_{q_\phi} [\nabla_\phi \log q_\phi(\boldsymbol{m} \mid \boldsymbol{a}) \cdot (\log p_\theta(\boldsymbol{a}, \boldsymbol{m}) - c)]$. The entropy term $\mathbb{H}[q_\phi(\boldsymbol{m}|\boldsymbol{a})]$ has a closed-form expression for DT-RPs as in Eq. (10), ensuring efficient optimization via SGD.

- **M-Step: Logic Model Parameter Update** Maximize the expected complete-data log-likelihood with respect to $\theta$:

$$\theta^{(t+1)} = \arg\max_\theta \mathbb{E}_{q_{\phi^{(t+1)}}(\boldsymbol{m}|\boldsymbol{a})} [\log p_\theta(\boldsymbol{a}, \boldsymbol{m})]$$

The expectation is approximated by sampling mental event trajectories $\boldsymbol{m} \sim q_{\phi^{(t+1)}}(\boldsymbol{m} \mid \boldsymbol{a})$.

**Comparison with VAEs** Unlike standard VAEs, which leverage the reparameterization trick for joint, differentiable updates of $\theta$ (decoder) and $\phi$ (encoder) (Kingma, 2013; Mehrasa et al., 2019), our framework employs staged E-M steps, alternating between freezing $\theta$ to optimize $\phi$ (E-step) and freezing $\phi$ to optimize $\theta$ (M-step). This modular approach mimics classical EM, reducing instability from competing parameter updates while accommodating rule-based refinement via column generation—a non-differentiable process incompatible with VAE-style gradients. To ensure stability, we run the E-step to near-convergence (ELBO change $< \epsilon$) before each M-step, anchoring updates in reliable posterior estimates.

## 4. Future Action Prediction

To predict future actions, we condition on both observed actions $\boldsymbol{a}$ and inferred mental states $\boldsymbol{m}$ sampled from the DT-RP posterior. The backtracking mechanism (as shown in Alg. 1) ensures temporal coherence by dynamically checking for mental state shifts between the last observed action and the predicted next action. If a mental event (e.g., *safety-check intent*) occurs in this interval, the sampler regenerates predictions using the updated mental state, aligning actions with the most recent intent.

## 5. Experiments

### 5.1. Experimental Setup

**Datasets** We conduct our experiments on synthetic and real-world datasets. For *synthetic datasets*, we simulate two datasets with same sample size (2000 sequences) and same time horizon (15s), but with different number of predicates and ground truth logic rules: *i) Syn Data-1*: 3 ground truth rules, 1 mental predicates and 2 action predicates. Each sequence has 18.60 actions on average, *ii) Syn Data-2*: a more complicated scenario with 4 ground truth rules, 2 mental predicates and 2 action predicates. Each sequence has 13.25

---

**Algorithm 1** Predicting Actions with Backtracking

1: **Step 1: Infer Mental States**
2: Given historical actions $\boldsymbol{a}$, sample mental events $\boldsymbol{m} \sim q_\phi(\boldsymbol{m} \mid \boldsymbol{a})$ over discrete time grids.
3: Define the augmented history: $\mathcal{H}(t) = \boldsymbol{a} \cup \boldsymbol{m}$.
4: **Step 2: Propose Candidate Action**
5: Generate a candidate action time and associated value: $t_{\text{next}}^a, x_{\text{next}}^a \sim p_\theta(\cdot \mid \mathcal{H}(t))$.
6: **Step 3: Check for Mental Events**
7: For the interval $[t_{\text{last}}, t_{\text{next}}^a)$, forward-sample mental events in discrete time:
8:     **if** a mental event occurs at $t^m \in [t_{\text{last}}, t_{\text{next}}^a)$:
9:       - Update the history by integrating the mental event at $t^m$.
10:       - Set $t_{\text{last}} = t^m$.
11:       - Regenerate $t_{\text{next}}^a, x_{\text{next}}^a \sim p_\theta(\cdot \mid \mathcal{H}(t^m))$.
12: **Step 4: Iterate with Maximum Rounds**
13: Repeat Steps 3 until no new mental events occur in $[t_{\text{last}}, t_{\text{next}}^a)$.
14: **Step 5: Accept $t_{\text{next}}^a, x_{\text{next}}^a$ as Final Predicted Action**

---

actions on average. For *real-world datasets*, we select four interesting datasets that capture human behaviors, which are highly likely to be driven by human mental states: *i) Hand-Me-That* (Wan et al., 2022): We focus on the *change-state* type episodes and extract 503 sequences with average action trajectory length 30.5. *ii) Car-following* (Li et al., 2023): We extract 2000 car-following behavior sequences with 3.6 average action events and average time horizon 19.44s. *iii) MultiTHUMOS* (Yeung et al., 2018): We focus only on the basketball dataset with 2000 sequences. The time horizon of each sequence is 208.32s with 38.41 actions on average. *iv) EPIC-Kitchen-100* (Damen et al., 2018): We focus on two goals: cut onion and pour water, and extract 131 sequences contains related key actions. The time horizon of each sequence is 500s with 5.41 actions on average. For all the datasets, we split each dataset into 80%, 10%, 10% train/dev/test by the total sequences. Detailed introduction about datasets can be found in Appendix. B

During the training process, only the action trajectories are given. For synthetic datasets, the ground truth mental events are known, allowing us to compare the inferred mental events with the ground truth. In real-world datasets, the ground truth mental events are hidden, and thus we cannot directly compare the accuracy of sampled mental events, but resort to compare the accuracy of action predictions.

**Baselines** We choose state-of-the-art baselines considering three different fields: *i) Neural Temporal Point Process Model*: RMTPP (Du et al., 2016), THP (Zuo et al., 2020), PromptTPP (Xue et al., 2023), and HYPRO (Xue et al., 2022), *ii) Logic-Based Model*: TELLER (Li et al., 2021)

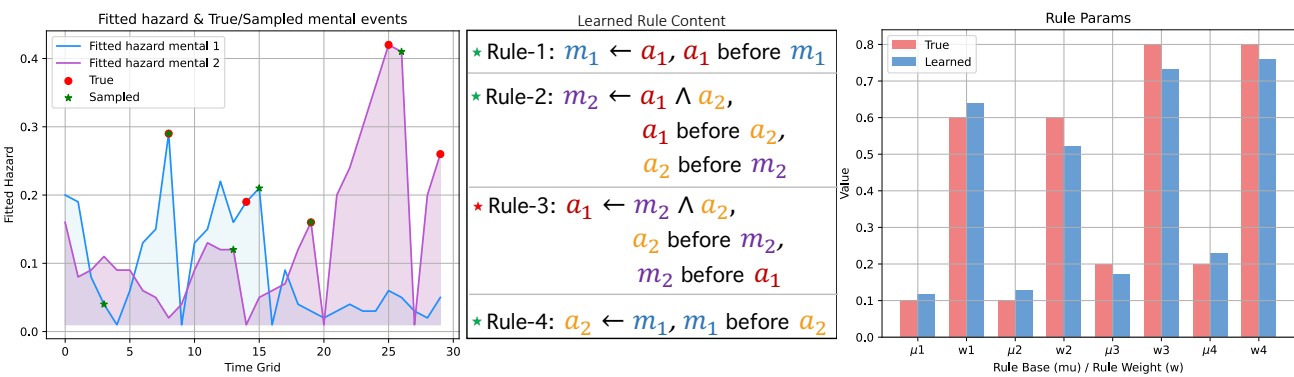

*Figure 3.* Results on Syn Data-2. **Left:** fitted hazards of mental events for one sequence, corresponding sampled mental events, and ground truth mental events. **Middle:** results of learned rules. Green stars indicate that the rules are correctly learned. **Right:** ground truth rule parameters and learned parameters.

| Category | Model | Syn Data-1 | | Syn Data-2 | | Hand-Me-That | |
|---|---|---|---|---|---|---|---|
| | | ER% ↓ | MAE ↓ | ER% ↓ | MAE ↓ | ER% ↓ | MAE ↓ |
| Neural TPP | RMTPP | $48.37 \pm 1.47$ | $3.11 \pm 0.09$ | $52.14 \pm 1.55$ | $3.20 \pm 0.07$ | $79.86 \pm 2.12$ | $2.13 \pm 0.10$ |
| | THP | $45.46 \pm 1.32$ | $2.83 \pm 0.08$ | $48.93 \pm 1.29$ | $2.99 \pm 0.07$ | $78.85 \pm 1.85$ | $2.08 \pm 0.12$ |
| | PromptTPP | $\underline{43.47 \pm 0.90}$ | $\underline{2.42 \pm 0.06}$ | $\underline{47.51 \pm 1.75}$ | $2.67 \pm 0.11$ | $76.35 \pm 2.06$ | $\underline{1.68 \pm 0.08}$ |
| | HYPRO | $44.53 \pm 1.57$ | $2.46 \pm 0.10$ | $47.85 \pm 1.62$ | $\underline{2.60 \pm 0.05}$ | $\underline{75.88 \pm 2.22}$ | $1.70 \pm 0.08$ |
| Rule-based Model | TELLER | $46.72 \pm 1.86$ | $2.64 \pm 0.15$ | $49.15 \pm 2.23$ | $3.04 \pm 0.17$ | $78.28 \pm 2.90$ | $1.86 \pm 0.14$ |
| | CLNN | $46.25 \pm 1.42$ | $2.57 \pm 0.12$ | $48.32 \pm 1.74$ | $2.82 \pm 0.13$ | $77.74 \pm 1.95$ | $1.82 \pm 0.07$ |
| | STLR | $45.05 \pm 1.76$ | $2.52 \pm 0.12$ | $48.23 \pm 1.83$ | $2.72 \pm 0.14$ | $77.25 \pm 1.81$ | $1.80 \pm 0.09$ |
| Gen. Model | AVAE | $45.13 \pm 0.93$ | $2.82 \pm 0.08$ | $47.53 \pm 1.62$ | $2.92 \pm 0.10$ | $80.12 \pm 1.75$ | $2.10 \pm 0.12$ |
| | GNTPP | $47.22 \pm 1.84$ | $2.97 \pm 0.17$ | $51.86 \pm 2.08$ | $3.19 \pm 0.16$ | $85.38 \pm 3.02$ | $2.69 \pm 0.12$ |
| | VEPP | $47.58 \pm 1.58$ | $3.01 \pm 0.04$ | $52.02 \pm 1.52$ | $3.22 \pm 0.13$ | $83.32 \pm 2.47$ | $2.51 \pm 0.11$ |
| | STVAE | $46.81 \pm 1.63$ | $2.76 \pm 0.07$ | $49.27 \pm 1.76$ | $3.02 \pm 0.10$ | $79.12 \pm 2.56$ | $2.17 \pm 0.09$ |
| – | **Ours\*** | $\mathbf{41.72 \pm 1.45}$ | $\mathbf{2.32 \pm 0.09}$ | $\mathbf{46.85 \pm 1.56}$ | $\mathbf{2.52 \pm 0.15}$ | $\mathbf{75.28 \pm 2.12}$ | $\mathbf{1.26 \pm 0.07}$ |

*Table 1.* Comparison between our model and baselines on all synthetic datasets and Hand-Me-That datasets for prediction tasks. Bold text represents the best result and underline denotes the second-best result. The performance is averaged over three different seeds and the standard deviation is stored after "±". Results from our model are shaded in red.

CLNN (Yan et al., 2023), STLR (Cao et al., 2024), *iii) Generative Model*: AVAE (Mehrasa et al., 2019), GNTPP (Lin et al., 2022), VEPP (Pan et al., 2020), and STVAE (Wang et al., 2023). For PromptTPP and HYPRO, we choose AttNHP (Yang et al., 2021) as their base model. For GNTPP, we choose the revised attentive history encoder and VAE probabilistic decoder. Details are in Appendix. C.

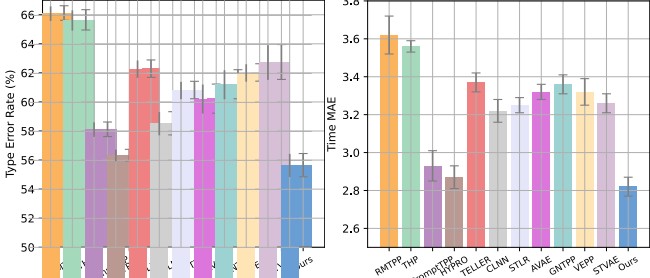

*Figure 4.* Performance of all the methods on predicting future 3 actions for Syn Data-2. **Left**: Comparison of event type average error rate ER%. **Right**: Comparison of event time average MAE.

**Comparison Metric** Following common next-event prediction task in TPPs (Du et al., 2016; Zuo et al., 2020; Yang et al., 2023), our model as well as other baselines attempt to predict the next event from history. Moreover, autoregressively predicting multiple future events is also considered in our experiments. We evaluate the event type prediction with the Error Rate (ER%) and evaluate the event time prediction with the Mean Absolute Error (MAE).

## 5.2. Experiments on Synthetic Dataset

**Infer Latent Mental Events and Learn Rule Parameters**
Results in Fig. 3 demonstrate a general alignment between the locations of fitted high-hazard and the actual occurrences of mental events. The mental events sampled based on these hazards also correspond reasonably well to the

| Category | Model | Car-Follow | | MultiTHUMOS | | EPIC-Kitchen | |
|---|---|---|---|---|---|---|---|
| | | ER%↓ | MAE↓ | ER%↓ | MAE↓ | ER%↓ | MAE↓ |
| Neural TPP | RMTPP | $35.71 \pm 1.33$ | $2.64 \pm 0.08$ | $67.01 \pm 2.64$ | $8.72 \pm 0.36$ | $49.02 \pm 2.38$ | $41.17 \pm 2.62$ |
| | THP | $33.43 \pm 1.62$ | $2.31 \pm 0.08$ | $62.32 \pm 2.34$ | $7.12 \pm 0.28$ | $42.19 \pm 2.36$ | $37.13 \pm 2.30$ |
| | PromptTPP | $33.29 \pm 0.87$ | $2.11 \pm 0.05$ | $60.35 \pm 1.92$ | $7.00 \pm 0.24$ | $\underline{40.82 \pm 1.80}$ | $\underline{33.21 \pm 1.78}$ |
| | HYPRO | $32.86 \pm 1.70$ | $\underline{2.03 \pm 0.10}$ | $\underline{58.25 \pm 2.42}$ | $\underline{6.98 \pm 0.37}$ | $42.28 \pm 2.02$ | $35.98 \pm 1.97$ |
| Rule-based Model | TELLER | $37.83 \pm 2.96$ | $3.41 \pm 0.22$ | $64.77 \pm 2.53$ | $7.52 \pm 0.48$ | $43.49 \pm 2,35$ | $38.05 \pm 3.02$ |
| | CLNN | $37.09 \pm 2.20$ | $3.25 \pm 0.18$ | $63.10 \pm 2.28$ | $7.33 \pm 0.34$ | $42.86 \pm 2.06$ | $37.13 \pm 1.89$ |
| | STLR | $\underline{32.75 \pm 1.90}$ | $2.47 \pm 0.12$ | $63.38 \pm 2.58$ | $7.69 \pm 0.32$ | $43.37 \pm 2.12$ | $36.85 \pm 1.79$ |
| Gen. Model | AVAE | $35.08 \pm 1.52$ | $2.95 \pm 0.13$ | $61.17 \pm 2.09$ | $8.32 \pm 0.42$ | $43.56 \pm 2.00$ | $39.24 \pm 2.18$ |
| | GNTPP | $39.22 \pm 1.77$ | $3.89 \pm 0.06$ | $63.75 \pm 2.30$ | $8.37 \pm 0.43$ | $46.25 \pm 2.17$ | $38.11 \pm 2.25$ |
| | VEPP | $40.25 \pm 2.39$ | $3.78 \pm 0.16$ | $64.23 \pm 2.63$ | $8.42 \pm 0.50$ | $47.56 \pm 2.42$ | $38.93 \pm 2.03$ |
| | STVAE | $37.23 \pm 1.78$ | $3.18 \pm 0.17$ | $64.28 \pm 2.64$ | $8.24 \pm 0.35$ | $45.83 \pm 2.09$ | $37.48 \pm 2.15$ |
| – | **Ours\*** | $\mathbf{32.72 \pm 2.90}$ | $\mathbf{1.80 \pm 0.18}$ | $\mathbf{57.20 \pm 2.32}$ | $\mathbf{6.76 \pm 0.45}$ | $\mathbf{40.26 \pm 2.20}$ | $\mathbf{32.19 \pm 2.24}$ |

*Table 2.* Comparison between our model and baselines on Car-Follow, MultiTHUMOS, and EPIC-Kitchen datasets for prediction tasks.

actual time points of occurrence. As we infer latent mental events through probability-based sampling, there is a certain degree of error involved. However, this error remains within an acceptable range. Our rule learning module correctly uncovers most ground truth rules and accurately learns the rule parameters. In particular, our model effectively addresses the challenge of limited data by incorporating logic rules as guidings. Even with a dataset size of only 2000 samples, it achieves promising results.

**Next Single Event Prediction** The experiments on two synthetic datasets to predict the next single future action events are presented in Tab. 1, from which one can observe that our model outperforms all the baselines.

**Next Multiple Events Prediction** Auto-regressive long-horizon prediction may cause *cascading error* in TPP (Xue et al., 2022). The HYPRO method considered in the baselines is purely data-driven, which is a flexible neural-based model combined with expressive energy-based models. In contrast, our method employs a neural black-box encoder, but with a rule-based white-box decoder. The prediction results rely on the learned logic rules. Our design choice inherently trades off interpretability for model expressiveness. Encouragingly, with enhanced interpretability, our model also mitigates cascading error. Shown in Fig. 4, in task of predicting next 3 actions, our model achieves comparable and even lower ER% and MAE than HYPRO.

### 5.3. Experiments on Real-World Dataset

**Experiment Results** On real-world datasets, we have designed prediction tasks that are specifically tailored to the characteristics of each dataset, taking into account variations in the number of future events to be predicted. For *Hand-Me-That*, *Car-Following*, and *EPIC-Kitchen-100* datasets, we focus on predicting the next action, whereas for *MultiTHU-*

*MOS* dataset, we aim to predict next 3 actions. The experimental results, presented in Tab. 1 and Tab. 2, demonstrate that our model performs exceptionally well in predicting both future event types and timings, outperforming all other methods. In practice, thanks to the inherent capabilities of our rule learning module, we can employ two strategies tailored to distinct scenarios: *autonomous rule learning for data-rich domains and template-guided learning for scenarios with limited data but ample prior knowledge*. In our experiments, we provide prior rule templates to enhance model performance as in Appendix. B. The rule learning module can refine these templates and learn rules from scratch. Indeed, our model uncovers overlooked rules with real-world significance, as in Appendix. D.4. In the Appendix. D.2, we also attempt to remove these prior knowledge. Despite this, our model still achieved satisfactory results in prediction tasks, showcasing its adaptability when applied to real-world datasets lacking prior knowledge.

**Prediction Examples** As exemplified in Fig. 9 for *Hand-Me-That* dataset and Fig. 10 for *Car-Follow* with analysis in Appendix. D.5, our model demonstrates intriguing applicability in real-life scenarios. Given our model's precise inference of historical human intentions and accurate forecasting of future human actions with corresponding time indexes, AI agents can promptly assist humans, significantly enhancing convenience.

### 5.4. Scalability, Ablation Study, and Hyper-Parameter

The scalability experiments on synthetic datasets shown in Fig. 5 validate that our model has potential to apply effectively to large-scale datasets. On real-world datasets, as discussed in Appendix. D.1, our model also demonstrates good scalability. The results of the ablation study on synthetic datasets are shown in Tab. 3, with detailed analysis can be found in Appendix. D.2, confirming that appropriate prior

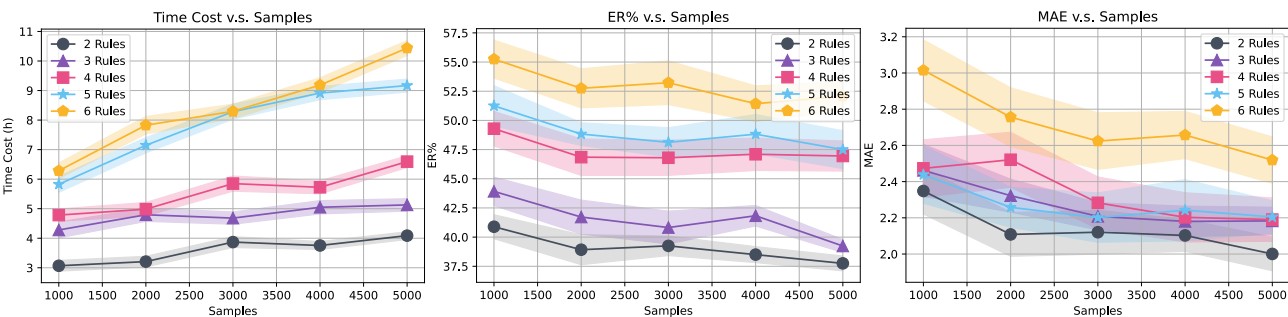

*Figure 5.* Scalability and the computation time cost of our model on synthetic datasets with varying sample size, ground truth temporal logic rules, and latent state domains. All the experiments are conducted over three random runs and the standard error is reflected in the shaded areas.

knowledge enhances the accuracy of model predictions, and the inclusion of backtracking indeed improves prediction accuracy, even when there exists noise in prior knowledge. Based on empirical results, our hyper-parameter selection, including time grid resolution and maximum number of backtracking rounds, strikes a balance between model performance and training efficiency, with experimental results reported in the Appendix. E.1.

| Ablation Settings | | Syn Data-1 | | Syn Data-2 | |
|---|---|---|---|---|---|
| Prior | Back | ER(%) | MAE | ER(%) | MAE |
| ✗ | ✗ | 48.28 | 3.13 | 52.50 | 3.24 |
| ✓ | ✗ | 45.64 | 2.64 | 48.37 | 2.86 |
| ✗ | ✓ | **41.72** | 2.32 | 46.85 | 2.52 |
| ✓ | ✓ | 42.08 | **2.19** | **46.43** | **2.47** |

*Table 3.* Ablation study on synthetic datasets. Our current selection of modules are highlighted in blue. "Prior" stands for prior knowledge and "Back" stands for backtracking mechanism

## 6. Related Work

**Neural Temporal Point Process (TPP)** Recent advancements in TPP field primarily concentrate on enhancing the flexibility of intensity functions. Some approaches utilized RNN and LSTM networks, such as (Du et al., 2016; Mei & Eisner, 2017; Xiao et al., 2017; Mei et al., 2020). Others leveraged Transformer architectures, including (Zuo et al., 2020; Zhang et al., 2020; Zhu et al., 2021; Yang et al., 2021). *Despite these advancements, the reliance on black-box models raises interpretability issues, particularly in contexts requiring explanations for events.* Recently, Xue et al. (2022) combined the base TPP with an energy function to overcome the cascading errors of predictions, *which is also what our model aims to avoid.*

**Logic-Informed Temporal Point Process** Logic rules effectively represent domain knowledge and hypotheses, offering explanations for real-world event data. A pioneering work (Li et al., 2020) introduced a unified framework integrating first-order logic rules to model event dynam-

ics through intensity of TPP. The follow-up work (Li et al., 2021) employed column generation for data-driven rule mining but faces scalability issues. Kuang et al. (2024) utilized EM algorithm to optimize the rule set in a differential manner. And the works of (Yan et al., 2023) and (Yang et al., 2024) developed a differentiable neuro-symbolic framework for modeling TPPs. *These works improve efficiency but overlook the importance of prior knowledge.* Zhang et al. (2021) incorporated neural TPP with temporal logic prior knowledge, achieving good performance. *In our paper, we customized logic-informed TPPs to model actions and mental events in continuous time. Using logic rules as priors, we enhance the inference of mental states and improve future action prediction, even with limited data.*

**Latent Mental Inference** Considering human intentions as latent variables, Wei et al. (2017) developed an EM-based approach for their inference but only model intentions as discrete latent variables. Mehrasa et al. (2019) and Zolotas & Demiris (2022) leveraged the VAE architecture to synthesize human trajectories, albeit without addressing the interpretability of the latent variables. Hidden Markov Models (HMMs) have also been employed for modeling sequential data and inferring hidden states. Jeong et al. (2021) integrated HMMs with VAEs to infer human activity sequences. The study by Cao et al. (2024) used learned spatial-temporal rules and human intentions to guide actions, most similar to our work. *However, it overlooks the time interval information between actions and mental events, where the elapsed time also carries valuable contextual insights.*

## 7. Conclusion

We propose a novel framework combining logic-informed TPPs with amortized variational EM to jointly infer latent mental states, learn the model parameters, and predict future actions. Guided by temporal logic rules, our model performs well even in small data scenarios and enhance interpretability. The introduction of the backtracking mechanism further improves the stability of our model on prediction tasks.

## Impact Statement

Our research proposes a novel framework combining logic-informed temporal point processes with amortized variational Expectation-Maximization to jointly infer unobserved mental states, learn the model parameters such as discrete rule content and continuous rule parameters, and predict future actions. Our method also holds practical significance. By inferring latent mental events and incorporating rule learning, our method enhances AI's ability to understand human thoughts and predict their actions, enabling timely assistance and promoting human-AI collaboration. The flexibility, adaptability, and interpretability showcase its potential to inspire future research endeavors.

## Acknowledgements

Shuang Li's research was in part supported by the Key Program of the NSFC under grant No. 72495131 , NSFC under grant No. 62206236, Shenzhen Stability Science Program 2023, Shenzhen Science and Technology Program ZDSYS20230626091302006, Longgang District Key Laboratory of Intelligent Digital Economy Security and SRIBD Innovation Fund SIF20240010.

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

## Appendix Overview

- Section A presents key additional technical details, including pseudocodes for each building block and the overall proposed amortized variational EM framework.

- Section B provides comprehensive explanation of the generating process of synthetic datasets, along with an overview of the real-world datasets and the corresponding pre-processing procedure. It also delineates the temporal logic rule templates as prior knowledge for real-world datasets.

- Section C elaborates on the baseline methods discussed in our paper.

- Section D shows more experimental details, including scalability experiments, ablation study, prediction examples on real-world datasets, et al.

- Section E provides reproducibility analysis, such as hyper-parameter selection and computing infrastructure.

- Section F states the limitation and broader impacts of our proposed method.

## A. Technique Details

### A.1. DT-RP Sampling

In Alg. 2, we provide the pseudocodes for sampling from discrete-time renewal processes.

---

**Algorithm 2** DT-RP Sampling for Event Type $x$

---

1: **Input:** Last event grid $k_0$, hazard function $h_x(\cdot)$
2: Sample $u \sim \text{Uniform}(0, 1)$
3: **for** $k = k_0 + 1, 2, \ldots$ **do**
4:     Compute $F_x(k) = 1 - \prod_{\tau=k_0+1}^{k} (1 - h_x(\tau))$
5:     **if** $F_x(k) > u$ **then**
6:         $t = k\Delta t$, update $k_0 \leftarrow k$    ▷ call "restart"
7:     **end if**
8: **end for**

---

### A.2. Adaptive Rule Learning via Column Generation

In Alg. 3, we provide the pseudocodes for adaptive rule learning module via column generation.

---

**Algorithm 3** Column Generation

---

1: **Initialize:** Start with $\mathcal{F}_0 = \emptyset$ or a set of predefined rules.
2: **while** not converged **do**
3:     **Step 1 (Master Problem):** Solve for $\tilde{\theta}_j^*$ given $\mathcal{F}_j$.
4:     **Step 2 (Sub Problem):** Generate candidates $\Delta\mathcal{F}$ using a heuristic or optimization strategy. Select $f^* = \arg\min_f \text{ReducedCost}(f)$.
5:     **Step 3:** Update $\mathcal{F}_{j+1} = \mathcal{F}_j \cup f^*$.
6: **end while**
7: **Terminate:** Stop when $\mathcal{F}_{j+1} = \mathcal{F}_j$ (reach the maximum iteration).

---

### A.3. Amortized Variational EM

In Alg. 4, we provide the overall training procedures of our proposed amortized variational EM framework.

### A.4. Backtracking Mechanism

In Alg. 5, we provide the pseudocodes for the backtracking sampling mechanism (we have reported in the main text).

---

**Algorithm 4** Integration with Variational EM

---

1: Within the M-step of the EM algorithm:

2:     **E-Step:** Freeze $\theta = \left[\tilde{\theta}, \mathcal{F}\right]$, optimize $\phi$ to infer $\boldsymbol{m} \sim q_\phi(\boldsymbol{m} \mid \boldsymbol{a})$.

3: **M-Step:** Alternates the following steps to obtain $\theta$:

- **Continuous Update:** Optimize $\tilde{\theta}$ via master problem.

- **Discrete Update:** Expand $\mathcal{F}_j$ via subproblem.

---

---

**Algorithm 5** Predicting Actions with Backtracking

---

1: **Step 1: Infer Mental States**

2: Given historical actions $\boldsymbol{a}$, sample mental events $\boldsymbol{m} \sim \boldsymbol{q}_\phi(\boldsymbol{m} \mid \boldsymbol{a})$ over discrete time grids.

3: Define the augmented history: $\mathcal{H}(t) = \boldsymbol{a} \cup \boldsymbol{m}$.

4: **Step 2: Propose Candidate Action**

5: Generate a candidate action time and associated value: $t^a_{\text{next}}, x^a_{\text{next}} \sim p_\theta(\cdot \mid \mathcal{H}(t))$.

6: **Step 3: Check for Mental Events**

7: For the interval $[t_{\text{last}}, t^a_{\text{next}})$, forward-sample mental events in discrete time:

8:     **if** a mental event occurs at $t^m \in [t_{\text{last}}, t^a_{\text{next}})$:

9:         - Update the history by integrating the mental event at $t^m$.

10:         - Set $t_{\text{last}} = t^m$.

11:         - Regenerate $t^a_{\text{next}}, x^a_{\text{next}} \sim p_\theta(\cdot \mid \mathcal{H}(t^m))$.

12: **Step 4: Iterate with Maximum Rounds**

13: Repeat Steps 3 until no new mental events occur in $[t_{\text{last}}, t^a_{\text{next}})$.

14: **Step 5: Accept $t^a_{\text{next}}, x^a_{\text{next}}$ as Final Predicted Action**

---

# B. Datasets and Prior Knowledge

## B.1. Datasets

- **Synthetic Dataset**

    - *Syn Data-1*: 3 ground truth rules, 1 mental predicates and 2 action predicates. Each sequence has 18.60 actions on average.

    - *Syn Data-2*: a more complicated scenario with 4 ground truth rules, 2 mental predicates and 2 action predicates. Each sequence has 13.25 actions on average.

- **Real-World Dataset**

    - *Hand-Me-That* (Wan et al., 2022): contains 10,000 episodes of human-robot interactions in household tasks with a textual interface. In each episode, the robot first observes a trajectory of human actions towards her internal goal. Next, the robot receives a human instruction and takes actions to accomplish the subgoal behind the instruction. Here we consider robot's actions as the expert human trajectory. We combine human's history trajectory and robot's subsequent actions as a whole sequence from a single agent (human) and transfer the intermediate human instruction into human's mental state (e.g., human instruction: *Please soak the piece of cloth on the toilet* can be regard as human's mental: *want to soak the cloth*). The 10,000 episodes can be classified based on 3 human's instruction types: *bring-me*, *move-to* and *change-state*. Considering the diversity and practicability of defined logic rule templates, we focus on involving more action and mental predicates instead of complex objects' names, we mainly use change-state episodes. Abandoning episodes without human history trajectories, we finally get 503 sequences with average length 30.5.

    - *Car-Following* (Li et al., 2023): is processed from Lyft level-5 open dataset. The Lyft level-5 dataset (Houston et al., 2021) is a large-scale dataset of high-resolution sensor data collected by a fleet of 20 self-driving cars. The dataset includes 1000+ hours of perception and motion data collected over a 4-month period from urban and suburban environments along a fixed route in Palo Alto, California. The dataset covers diverse Car-Following (CF) regimes and the enhanced dataset provides smooth, ready-to-use motion information for Car-Following behaviors

investigation. A regime refers to a driving situation experienced by the following vehicle (usually restricted by the leading vehicle). 29k+ Human Vehicle (HV)-following-Autonomous Vehicle (AV) pairs and 42k+ Human Vehicle (HV)-following-Human Vehicle (HV) pairs were selected and enhanced in similar environments from the Lyft level-5 dataset, with the total duration spanning over 460+ hours, covering a total distance of 15,000+ km. We mainly focus on HV-following-HV CF pairs because the essential information of HV-following-AV CF pairs (AV's speed and acceleration which are used to segment vehicle's regimes) were estimated by Kalman filtering, while all the information in HV-following-HV CF pairs are truly recorded with slight imputation of missing data. We extracted 2000 sequences from HV-following-HV CF pairs and defined 3 reasonable logic rule templates to explain the change of regimes in those sequences.

– *MultiTHUMOS* (Yeung et al., 2018): a challenging dataset for action recognition, containing 400 videos of 65 different human actions. In this paper, we focus only on the basketball dataset with 2000 sequences. The time horizon of each sequence is 208.32s with 38.41 actions on average.

– *EPIC-Kitchen-100* (Damen et al., 2018): a large-scale dataset in first-person (egocentric) vision, which are multi-faceted, audio-visual, non-scripted recordings in native environments - i.e. the wearers' homes, capturing all daily activities in the kitchen over multiple days. In this paper, we focus on two goals in the kitchen: cut onion and pour water, and extract 131 sequences contains related key actions. The time horizon of each sequence is 500s with 5.41 actions on average.

### B.2. Prior Knowledge

For synthetic datasets, we know the ground truth temporal logic rules so that we can compare the rule learning accuracy. For real-world datasets, we have defined a set of temporal logic rule templates as prior knowledge that align with intuition and experiential knowledge, capturing the time-based patterns associated with human mental intentions. The prior knowledge temporal logic rule templates are partial and may not entirely correct but potentially capture some patterns of the ground truth rules. Our model aims to base on these kinds of prior knowledge to refine and generate more accurate logic rules on real-world dataset. It is noteworthy that these human-mental-related predicates are latent and do not actually exist within these real-world datasets.

| Predicates | Explanation |
|:---:|:---:|
| $m_1$ | mental event-1 |
| $a_1$ | action event-1 |
| $a_2$ | action event-2 |

*Table 4.* Defined predicates and corresponding explanation for Syn data-1.

| Rule Num | Rule Content | Rule Weight |
|:---:|:---:|:---:|
| Rule-1 | $m_1 \leftarrow a_1, (a_1$ before $m_1)$ | 0.6 |
| Rule-2 | $a_1 \leftarrow a_2, (a_2$ before $a_1)$ | 0.6 |
| Rule-3 | $a_2 \leftarrow m_1, (m_1$ before $a_2)$ | 0.8 |

*Table 5.* Ground truth temporal logic rules and corresponding weights for Syn Data-1.

| Predicates | Explanation |
|:---:|:---:|
| $m_1$ | mental event-1 |
| $m_2$ | mental event-2 |
| $a_1$ | action event-1 |
| $a_2$ | action event-2 |

*Table 6.* Defined predicates and corresponding explanation for Syn data-2.

- **Synthetic Dataset**

    – *Syn Data-1*: Defined predicates and ground truth temporal logic rules are shown in Tab. 4 and Tab. 5 respectively. We can compare the rule learning accuracy based on synthetic dataset, since we know the ground truth.

    – *Syn Data-2*: Defined predicates and ground truth temporal logic rules are shown in Tab. 6 and Tab. 7 respectively.

| Rule Num | Rule Content | Rule Weight |
|----------|--------------|-------------|
| Rule-1 | $m_1 \leftarrow a_1, (a_1 \text{ before } m_1)$ | 0.6 |
| Rule-2 | $m_2 \leftarrow a_1 \wedge a_2, (a_1 \text{ before } a_2), (a_2 \text{ before } m_2)$ | 0.6 |
| Rule-3 | $a_1 \leftarrow m_2 \wedge a_2, (m_2 \text{ before } a_2), (a_2 \text{ before } a_1)$ | 0.8 |
| Rule-4 | $a_2 \leftarrow m_1, (m_1 \text{ before } a_2)$ | 0.8 |

*Table 7.* Ground truth temporal logic rules and corresponding weights for Syn Data-2.

- **Real-World Dataset**

  - *Hand-Me-That*: Extracted predicates and prior knowledge temporal logic rule templates are shown in Tab. 8 and Tab. 9 respectively.
  - *Car-Following*: Extracted predicates and prior knowledge temporal logic rule templates are shown in Tab. 10 and Tab. 11 respectively.
  - *MultiTHUMOS*: Extracted predicates and prior knowledge temporal logic rule templates in Tab. 12 and Tab. 13 respectively
  - *EPIC-Kitchen-100*: Extracted predicates and prior knowledge temporal logic rule templates are defined in Tab. 14 and Tab. 15 respectively.

| Predicates | Explanation |
|------------|-------------|
| MoveTo | Move to a location or an object |
| PickUp | Pick up an object from a location or a receptacle |
| Put | Put an object on a location or into a receptacle |
| ToggleOn | Toggle on toggleable-thing, like electric device |
| Soak | Soak an object |
| Open | Open openable thing, like cabinet |
| Clean | Clean an object or a location |
| Cool | Freeze food |
| Slice | Slice food |
| Heat | Heat food |
| Close | Close openable thing |
| WantToPickUp | Want to get an object |
| WantToSoak | Want to soak an object |
| WantToOpenToGet | Want to open an openable thing to get an object |
| WantToToggleOn | Want to toggle on an electric device |
| WantToPut | Want to put an object on a location or into a receptacle |
| WantToClean | Want to clean a location or an object |
| WantToHeat | Want to heat food |
| WantToCool | Want to freeze food |
| WantToSlice | Want to slice an object |

*Table 8.* Defined predicates and corresponding explanation for Hand-Me-That dataset.

## C. Baselines

In this paper, we primarily focus on baselines from three different fields: neural temporal point process model, logic-based model, and generative model. Below, we provide a detailed introduction to these baselines.

- **Neural Temporal Point Process Model**

  - RMTPP (Du et al., 2016): The approach considers the intensity function of a temporal point process as a nonlinear function that depends on the history. It utilizes a recurrent neural network to automatically learn a representation of the influences from the event history, which includes past events and time intervals, thereby fitting the intensity function of the temporal point process.

| Rule Num | Rule Content |
|---|---|
| Rule-1 | PickUp ← WantToPickUp ∧ MoveTo, (WantToPickUp before MoveTo), (MoveTo before PickUp) |
| Rule-2 | Soak ← WantToSoak ∧ MoveTo ∧ Put ∧ ToggleOn, (WantToSoak equal MoveTo), (MoveTo before Put), (Put before ToggleOn), (ToggleOn before Soak) |
| Rule-3 | PickUp ← WantToOpenToGet ∧ MoveTo ∧ Open ∧ (WantToOpenToGet before MoveTo), (MoveTo before Open), (Open before MoveTo) |
| Rule-4 | ToggleOn ← WantToToggleOn ∧ MoveTo, (WantToToggleOn equal MoveTo), (MoveTo before ToggleOn) |
| Rule-5 | Put ← WantToPut ∧ MoveTo, (WantToPut equal MoveTo), (MoveTo before Put) |
| Rule-6 | Clean ← WantToClean ∧ Soak ∧ PickUp ∧ MoveTo, (WantToClean before Soak), (Soak before PickUp), (PickUp before MoveTo), (MoveTo before CLean) |
| Rule-7 | Heat ← WantToHeat ∧ PickUp ∧ MoveTo ∧ Put ∧ ToggleOn, (WantToHeat before PickUp), (PickUp before MoveTo), (MoveTo before Put), (Put before ToggleOn), (ToggleOn before Heat) |
| Rule-8 | Cool ← WantToCool ∧ PickUp ∧ MoveTo ∧ Open ∧ Put ∧ Close, (WantToCool before PickUp), (PickUp before MoveTo), (MoveTo before Open), (Open before Put), (Put before Close), (Close before Cool) |
| Rule-9 | Slice ← WantToSlice ∧ Put ∧ PickUp, (WantToSlice before Put), (Put before PickUp), (PickUp before Slice) |

*Table 9.* Temporal logic rules as prior knowledge for Hand-Me-That dataset.

| Predicates | Explanation |
|---|---|
| Fa | Free acceleration |
| C | Cruising at a desired speed |
| A | Acceleration following a leading vehicle |
| D | Deceleration following a leading vehicle |
| F | Constant speed following |
| ConservativeIntention | The driver has a conservative intention, maintaining their speed |
| AggressiveIntention | The driver has an aggressive intention, tending to accelerate |

*Table 10.* Defined following car predicates and corresponding explanation in Car-Following dataset.

– THP (Zuo et al., 2020): The model employs a concurrent self-attention module to embed historical events and generate hidden representations for discrete time stamps. These hidden representations are then used to model the interpolated continuous time intensity function. THP can also incorporate additional structural knowledge. Importantly, THP surpasses RNN-based approaches in terms of computational efficiency and the ability to capture long-term dependencies.

– PromptTPP (Xue et al., 2023): The model incorporates a continuous-time retrieval prompt pool into the base TPP, enabling sequential learning of event streams without the need for buffering past examples or task-specific attributes. Specifically, this approach consists of a base TPP model, a pool of continuous-time retrieval prompts, and a prompt-event interaction layer. By addressing the challenges associated with modeling streaming event sequences, this mode enhances the model's performance.

– HYPRO (Xue et al., 2022): The hybridly normalized probabilistic (HYPRO) model is capable of making long-horizon predictions for event sequences. This model consists of two modules: the first module is an auto-regressive

| Rule Num | Rule Content |
|---|---|
| Rule-1 | A ← C ∧ AggressiveIntention, (C before AggressiveIntention), (AggressiveIntention equal A) |
| Rule-2 | C ← Fa ∧ ConservativeIntention, (Fa before ConservativeIntention) (ConservativeIntention equal C) |
| Rule-3 | F ← A ∧ D ∧ ConservativeIntention, (A before D), (D before ConservativeIntention), (ConservativeIntention before F) |

*Table 11.* Temporal logic rules as prior knowledge for Car-Following dataset.

| Predicates | Explanation |
|---|---|
| Dribble | Dribbling the basketball |
| Pass | Passing the basketball from one person to another |
| Shot | An attempt to put the basketball in the basketball hoop |
| PoorShootingOpportunity | Mental event, a player think that this is not a good shooting opportunity |
| GoodShootingOpportunity | Mental event, a player think that this is a good shooting opportunity |

*Table 12.* Defined predicates and corresponding explanation for MultiTHUMOS basketball dataset.

| Rule num | Rule Content |
|---|---|
| Rule-1 | Dribble ← PoorShootingOpportunity, (PoorShootingOpportunity before Dribble) |
| Rule-2 | Pass ← PoorShootingOpportunity, (PoorShootingOpportunity before Pass) |
| Rule-3 | Shot ← GoodShootingOpportunity, (GoodShootingOpportunity before Shot) |

*Table 13.* Temporal logic rules as prior knowledge for MultiTHUMOS basketball dataset.

base TPP model that generates prediction proposals, while the second module is an energy function that assigns weights to the proposals, prioritizing more realistic predictions with higher probabilities. This design effectively mitigates the cascading errors commonly experienced by auto-regressive TPP models in prediction tasks, thereby improving the model's accuracy in long-term forecasting.

- **Logic-Based Model**

  - TELLER (Li et al., 2021): It is a non-differentiable algorithm that can be described as a temporal logic rule learning algorithm based on column generation principles. This method formulates the process of discovering rules from noisy event data as a maximum likelihood problem. It also designs a tractable branch-and-price algorithm to systematically search for new rules and expand existing ones. The algorithm alternates between a rule generation stage and a rule evaluation stage, gradually uncovering the most significant set of logic rules within a predefined time limit.

  - CLNN (Yan et al., 2023): The model learns weighted clock logic (wCL) formulas, which serve as interpretable temporal logic rules indicating how certain events can promote or inhibit others. Specifically, the CLNN model captures temporal relations between events through conditional intensity rates guided by a set of wCL formulas that offer greater expressiveness. In contrast to conventional approaches that rely on computationally expensive combinatorial optimization to search for generative rules, CLNN employs smooth activation functions for the components of wCL formulas. This enables a continuous relaxation of the discrete search space and facilitates efficient learning of wCL formulas using gradient-based methods.

  - STLR (Cao et al., 2024): A model specifically designed for learning spatial-temporal logic rules in order to explain human actions. It consists of two main modules: the rule generator, employing the transformer to infer logic rules by treating them as latent variables, and the reasoning evaluator, which predicts future entity trajectories based on the generated rules. While STLR demonstrates flexibility in generating logic rules without relying on prior knowledge, it lacks the ability to infer fine-grained latent mental events in a real-time manner, a capability inherent in our method.

- **Generative Model**

| Predicates | Explanation |
|---|---|
| TakePlate | Retrieve the plate for future use |
| TakeEggs | Retrieve the eggs for further use |
| TakeOnion | Retrieve the onion for further use |
| TakeGlass | Retrieve the glass for further use |
| CutOnion | Retrieve the onion for further use |
| PourWater | Pour water to glass |
| NeedOnionToCook | Mental event, one has the intention to use onion to cook |
| NeedWater | Mental event, one needs water in kitchen |

*Table 14.* Defined predicates and corresponding explanation for EPIC-Kitchen-100 dataset.

| Rule num | Rule Content |
|----------|--------------|
| Rule-1 | CutOnion ← NeedOnionToCook ∧ TakeOnion, (NeedOnionToCook before TakeOnion), (TakeOnion before CutOnion) |
| Rule-2 | PourWater ← NeedWater ∧ TakeGlass, (NeedWater before TakeGlass), (TakeGlass before PourWater) |

*Table 15.* Temporal logic rules as prior knowledge for EPIC-Kitchen-100 dataset.

– AVAE (Mehrasa et al., 2019): The model is a recurrent variational auto-encoder designed for modeling asynchronous action sequences. At each time step, the model utilizes the history of actions and inter-arrival times to generate a distribution over latent variables. A sample from this distribution is then decoded into probability distributions for the inter-arrival time and action label of the next action. To address the limitations of using a fixed prior in the traditional VAE framework, this model incorporates a prior net that enhances the learning process.

– GNTPP (Lin et al., 2022): The model is a comprehensive generative framework for neural temporal point process modeling. It utilizes deep generative models as probabilistic decoders to approximate the target distribution of occurrence time. For the encoder, the model considers both RNN-based methods and self-attention-based mechanisms. As for the decoder, the model incorporates multiple generative models, such as the temporal conditional diffusion denoising model, temporal conditional VAE model, temporal conditional GAN model, temporal conditional continuous normalizing flow model, and temporal conditional noise score network model. The various combinations of encoders and decoders make the GNTPP highly flexible.

– VEPP (Pan et al., 2020): This model is a probabilistic generative model for event sequences. It introduces VAE to event sequence modeling that can better use the latent information and capture the distribution over inter-arrival time and types of event sequences. Specifically, it employs LSTM to embed the event sequence and utilizes VAE for modeling the event sequence.

– STVAE (Wang et al., 2023): This model combines the classical temporal point process with the neural variational inference framework, leading to its good ability to model human trajectories with continuous temporal distribution, variable length, and multi-dimensional context information.

## D. Experimental Details

### D.1. Scalability

To test the scalability of our proposed model, we have varied the training sample size within $\{1000, 2000, 3000, 4000, 5000\}$ as well as the ground truth rules within $\{2, 3, 4, 5, 6\}$ for synthetic datasets. Synthetic datasets with complex ground truth rules have larger corresponding domains for latent mental states. For example, synthetic datasets with 5 ground truth rules typically have latent mental state domains encompassing 4 mental states, ensuring that each of these 4 mental states is associated with at least one ground truth rule. Synthetic datasets with 6 ground truth rules are the most challenging with latent mental state domains encompassing 6 mental states. We have also extracted more data sequences for Hand-Me-That and Car-Following datasets to investigate the scalability of our model to handle large-scale real-world datasets.

As depicted in Fig. 6 and Fig. 7, our model swiftly converges with acceptable training time even with large-scale datasets for synthetic dataset and real-world dataset. The model's prediction performance improves with larger sample sizes, demonstrating its good scalability.

For more intricate rules and larger mental space domains like dataset with 6 ground truth rules (colored in orange in Fig. 6), the prediction ER% decreased to 52.08% and the MAE reduced to 2.52 when provided with 5000 samples, highlighting its ability to handle complex domains.

For these two real-world datasets with different sample sizes (results are in Fig. 7), the prediction performance improved with more samples and also resulted in more computational cost. For small-scale real-world dataset, our model well-handles challenges posed by small datasets. Even with a dataset size of only few samples, our model delivers satisfactory results. For the Hand-Me-That dataset with 100 samples, the ER% and MAE are 78.33% and 1.32. For the Car-Following dataset with 1000 samples, the ER% and MAE are 35.38% and 2.12, which are comparable with the majority baselines trained with larger sample size. For large-scale real-world dataset, the ER% and MAE decrease to 72.09% and 1.24 for Hand-Me-That dataset with 1000 samples. And these two metrics decrease to 30.18% and 1.69 for the Car-Following dataset with 5000 samples. Even on large-scale datasets, our algorithm converges relatively quickly with current computational infrastructure,

indicating the ability of our proposed model to handle large-scale real-world datasets. This is attributed to the constraints of rule length to reduce search space, thereby mitigating computational complexity to some extent.

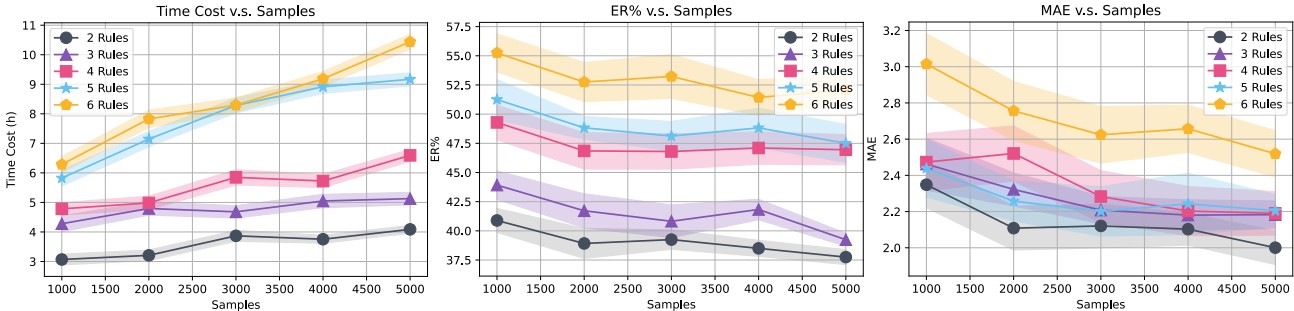

*Figure 6.* Scalability and the computation time cost of our model on synthetic dataset with varying sample size, ground truth temporal logic rules, and latent state domains. All the experiments are conducted over three random runs and the standard error is reflected in the shaded areas.

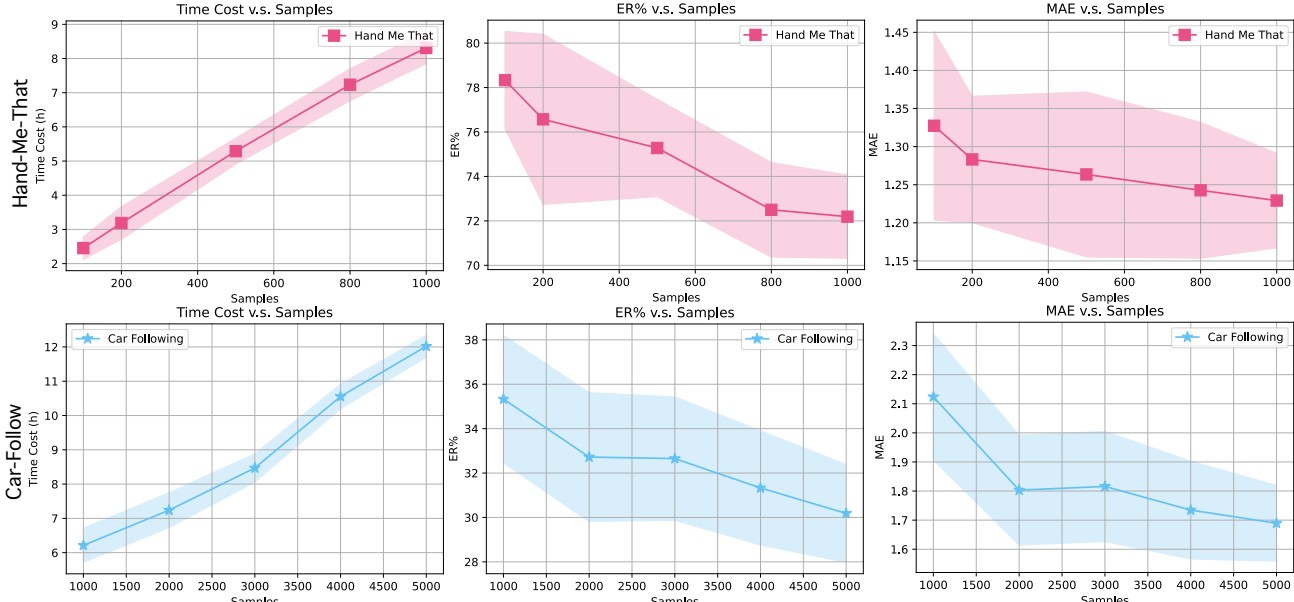

*Figure 7.* Scalability and the computation time cost of our model on real-world datasets (Hand-Me-That and Car-Following datasets) with varying sample size. All the experiments are conducted over three random runs and the standard error is reflected in the shaded areas.

## D.2. Ablation Study

We conducted an ablation study to assess the importance of different components, using the following ablation settings: (i) removing the prior knowledge and removing the backtracking module, (ii) solely removing the backtracking module, (iii) solely removing the prior knowledge module, (iv) evaluating the full model. The results on synthetic datasets and real-world datasets are shown in Tab. 16 and Tab. 17.

Note that if we exclude the prior knowledge module, the rule learning module will commence with an empty rule set. Consequently, the training process will require additional time to converge due to increased iterations between solving the master problem and solving the sub problems within the rule learning module. The experiments were conducted with consistent settings and hyper-parameters as described in our paper.

For synthetic dataset, the results from different ablation settings confirm that appropriate prior knowledge enhances the accuracy of model predictions. Even though, our model without prior knowledge still perform well on prediction tasks on two synthetic datasets, demonstrating the capability of our model being adapt to situations lacking prior knowledge. Additionally, the inclusion of a backtracking mechanism plays a more vital role which significantly improves model

| Ablation Settings | | Syn Data-1 | | Syn Data-2 | |
|---|---|---|---|---|---|
| Prior Knowledge | Backtracking | ER(%) | MAE | ER(%) | MAE |
| ✗ | ✗ | 48.28 | 3.13 | 52.50 | 3.24 |
| ✓ | ✗ | 45.64 | 2.64 | 48.37 | 2.86 |
| ✗ | ✓ | **41.72** | 2.32 | 46.85 | 2.52 |
| ✓ | ✓ | 42.08 | **2.19** | **46.43** | **2.47** |

*Table 16.* Ablation study on synthetic datasets. Our current selection of modules are highlighted in blue.

| Ablation Settings | | Hand-Me-That | | Car-Follow | | MultiTHUMOS | | EPIC-Kitchen | |
|---|---|---|---|---|---|---|---|---|---|
| Prior Knowledge | Backtracking | ER(%) | MAE | ER(%) | MAE | ER(%) | MAE | ER(%) | MAE |
| ✗ | ✗ | 82.20 | 2.48 | 36.23 | 2.52 | 65.41 | 7.29 | 47.52 | 42.68 |
| ✓ | ✗ | 79.45 | 1.72 | 35.88 | 2.49 | 67.26 | 7.47 | 46.82 | 39.51 |
| ✗ | ✓ | 77.14 | 1.38 | 34.60 | 1.92 | 58.10 | **6.52** | **40.15** | 33.23 |
| ✓ | ✓ | **75.28** | **1.26** | **32.72** | **1.80** | **57.20** | 6.76 | 40.26 | **32.19** |

*Table 17.* Ablation study on real-world datasets. Our current selection of modules are highlighted in blue.

performance, even when there is some level of noise in the prior knowledge. Overall, both modules contribute to enhancing the model's efficiency.

We want to emphasize that the practical significance of the ablation study experiments on real-world datasets, as obtaining prior knowledge in real data settings is sometimes challenging. The ability of our model to perform well in scenarios lacking prior knowledge is a crucial aspect of assessing its performance. Encouragingly, even after removing prior rules and starting with an empty rule set, our model still achieved good predictive performance by learning rules from scratch using the proposed rule learning module.

Furthermore, when providing the model with prior rules or rule templates, there was an improvement in predictive accuracy on real-world datasets. It is noteworthy that these prior logic rules are simple, readily available, and align with human cognitive logic. This indicates that our model effectively leverages prior knowledge to enhance performance while demonstrating the capability to handle datasets lacking such prior knowledge.

Overall, in practice, we can employ two strategies tailored to distinct scenarios: autonomous rule learning for data-rich domains and template-guided learning for scenarios with limited data but ample prior knowledge. These approaches enhance the adaptability of our model when applied to real-world datasets.

### D.3. Illustration of Attention Weight for Observed Actions

In our proposed model, the temporal point processes involve triggering between latent mental and action events, where historical actions can influence latent events and vice versa. Therefore, we resort to the attention mechanism to map the information of the entire action sequence on the spanning time horizon on each discrete time grid. Note that the attention in our model cannot capture how the mental state influences actions. The influence of mental process on action process is reflected on the intensity function after the inference of latent mental process.

In Fig. 8, we provide an example of attention weights for action sequence of Syn Data-1 on different discrete time grids, which visualizes attention patterns of different attention heads. Pixel $(i, k)$ in each figure signifies the attention weight of the event $(t_i^a, x_i^a)$ attending to the discrete time grid $k$. We can see that each attention head employs a different pattern to capture dependencies. For each attention head, the impact of one event is different on each discrete time grid, reflected by various attention weights. The impact of the entire action sequence on a discrete time grid can be conceptualized as a weighted combination of the entire action sequence, with weights derived through attention mechanisms. The attention mechanism effectively captures the potential influence of historical events at specific time points on the intensity of future events.

### D.4. Learned Rules with Real-World Significance on Real-World Dataset

In Tab. 18, we report several temporal logic rules that identified with real-world significance on real-world datasets, which are learned by our rule learning modules. These rules are not provided as prior knowledge and are easily overlooked. For *Hand-Me-That*, our rule learning model mines that after and intention *WantToPickUp*, human may sequentially take actions *MoveTo* and *Open*, thus increase the likelihood of taking action of *PickUp*. This rule is overlooked in our prior knowledge,

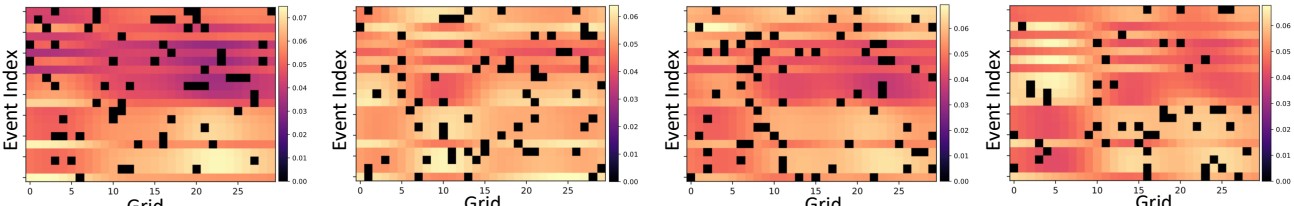

*Figure 8.* Visualization of attention patterns of different attention heads for action sequences in Syn Data-1 on different discrete time grids.

but aligns with human behavioral logic. The learned rule on *MultiTHUMO* dataset is representative, as a basketball player dribbles to seek offensive opportunities, once they identify a favorable moment, they take the shot. Our learned logic rules effectively capture this behavior on the court. Although we did not provide such prior knowledge during model training, our model still identified this logic rule adaptively through column generation in a data-driven manner. Other interesting rules found for other two real-world datasets also hold practical meaning, showcasing the good adaptability and the effectiveness of our rule learning module.

| Real-World Dataset | Rule Content |
|---|---|
| Hand-Me-That | PickUp ← WantToPickUp ∧ MoveTo ∧ Open, (WantToPickUp before MoveTo), (MoveTo before Open), (Open before PickUp) |
| Car-Follow | A ← C ∧ AggressiveIntention ∧ Fa, (C before AggressiveIntention), (AggressiveIntention equal Fa), (Fa before A) |
| MultiTHUMOS | Shot ← Dribble ∧ GoodShootingOpportunity, (Dribble before GoodShootingOpportunity), (GoodShootingOpportunity before shot) |
| EPIC-Kitchen | Need Water ← NeedOnionToCook ∧ TakeOnion, (NeedOnionToCook before TakeOnion), (TakeOnion before Need Water) |

*Table 18.* Learned temporal logic rules (we only report some of them) with real-world significance on real-world datasets.

## D.5. Prediction Example on Real-World Dataset

Our model demonstrates intriguing applicability in real-life scenarios due to its ability of accurately predicting real-world events and speculating on human thoughts. As exemplified by the *Hand-Me-That* dataset, in Fig. 9, our proposed model effectively infers human historical intentions like *want to clean the grill*, and *want to soak*. It also forecasts future human actions with correct time indexes. In this instance, if the AI-Agent infers a person's intention to clean the grill at time index 20 and predicts that the person will clean the grill at time index 22, it can promptly retrieves the grill for him, which will significantly enhance convenience.

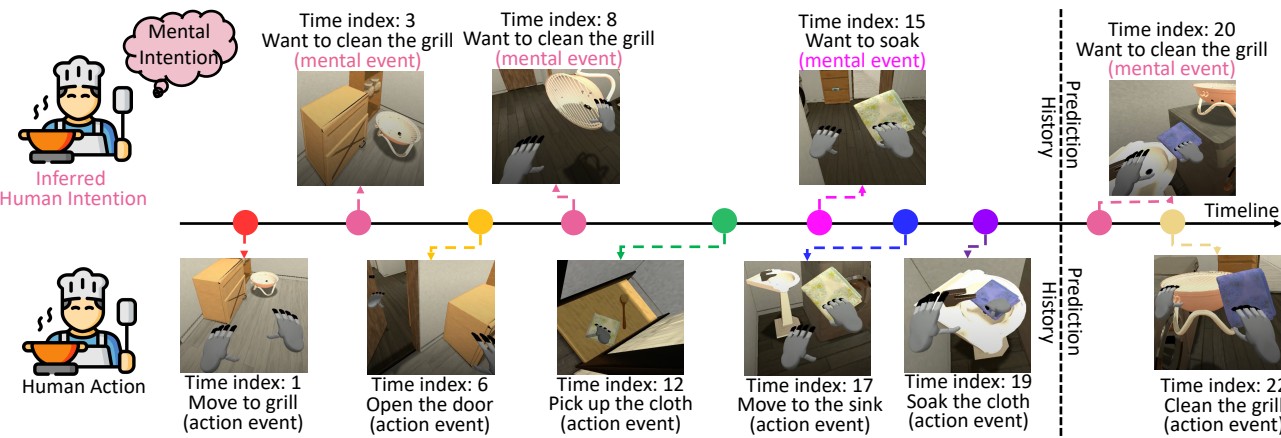

*Figure 9.* Inferred history mental events and predicted future events for one human trajectory aiming to clean the grill. **Top**: inferred and predicted human mental events. **Bottom**: observed and predicted human actions. Notice that the original dataset only includes the order of action occurrences. During data pre-processing, we assume actions to have equal time intervals of 1 and then discretize the timeline. Then, we can use time indices to represent specific time point.

We also provide another prediction example for *Car-Following* dataset. In Fig. 10, our model infers a driver's historical intentions and predicts their future car-following actions. In the field of autonomous driving, a self-driving vehicle can adjust its lane and drive reasonably by considering the inferred intentions of human drivers in neighboring lanes and their predicted behaviors, all while adhering to traffic regulations.

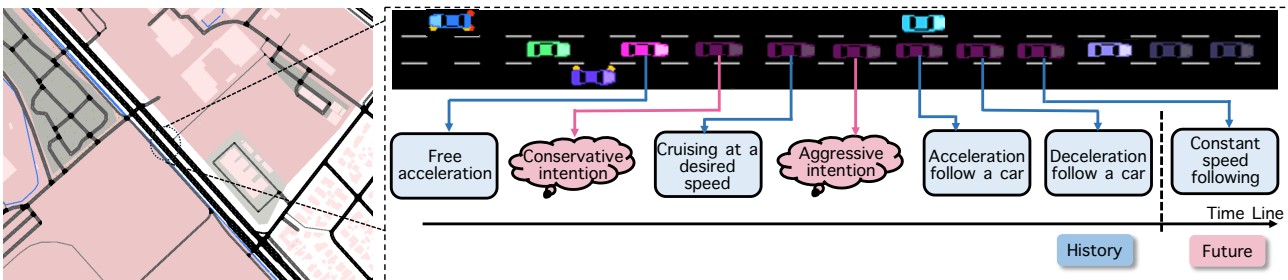

*Figure 10.* **Left:** Satellite map of Palo Alto, California, extracted from Google Earth, **Right:** Car following process (pink car) for one car trajectory. The historical mental event inferred by our method is indicated within the pink boxes. The next action in the future of the pink car predicted by our method is represented in the blue box to the right of the dashed line. The visualization is enhanced via SUMO simulator (Krajzewicz et al., 2002; Song et al., 2014).

## E. Reproducibility Analysis

### E.1. Hyper-Parameter Selection

We present the selected hyper-parameters on both synthetic datasets and real-world datasets in Tab. 19. The hyper-parameter selection metrics reflect the trade-off between training converged log-likelihood, prediction performance, and time efficiency.

| Hyper-parameters | Value Used | | | | | |
|---|---|---|---|---|---|---|
| | Syn Data-1 | Syn Data-2 | Hand-Me-That | Car-Following | MultiTHUMOS | EPIC-Kitchen |
| Batch Size | 32 | 32 | 32 | 64 | 64 | 16 |
| Time Grid Resolution | 0.50 | 0.50 | 1.00 | 0.25 | 1.50 | 2.50 |
| Max. Backtracking Rounds | 3 | 3 | 2 | 2 | 5 | 3 |
| Embedding Dimension | 32 | 32 | 16 | 32 | 32 | 16 |
| Dropout | 0.15 | 0.15 | 0.10 | 0.15 | 0.15 | 0.10 |
| Learning Rate | 1e-3 | 1e-3 | 5e-4 | 1e-3 | 1e-3 | 5e-4 |

*Table 19.* Descriptions and values of hyper-parameters used for models trained on both synthetic dataset and real-world datasets.

Specifically, the selection of some key hyper-parameters such as time grid resolution and maximum number of backtracking rounds is based on empirical results, which strikes a balance between model performance and training efficiency. We use metrics including training ELBO, training time, and prediction accuracy to select appropriate resolution of a discretized time grid and the number of backtracking rounds, with experiment results shown in Tab. 20 and Tab. 21.

For the resolution of time grid, as shown in Tab. 20, from the results one can see that under current selection of 0.50, the proposed model achieves the second highest ELBO on both two synthetic datasets. Reducing the resolution to 0.25 only marginally increases the ELBO to 18.45 on Syn Data-1 and 22.08 on Syn Data-2, but extends the model convergence time by 1.78 hours on Syn Data-1 and 1.84 hours on Syn Data-2 respectively. Conversely, while increasing the resolution can only slightly shorten training time. Moreover, it also decreases the model's converged ELBO and results in a decline in prediction accuracy when using the well-trained model for prediction tasks. This could be due to missing some crucial latent mental events. Hence, balancing between accuracy and computational efficiency, we opt for the current resolution of 0.5.

For the maximum number of backtracking rounds, as shown in Tab. 21, under our current selection of up to 3 rounds of backtracking, satisfactory results have been achieved in prediction tasks. Increasing the maximum backtracking rounds to 5 shows minimal improvement in prediction accuracy while adding to computational complexity. Conversely, reducing the backtracking rounds to a maximum of 2 or fewer significantly decreases prediction accuracy. This pattern is consistent across both synthetic datasets.

For real-world datasets, the selection of key hyper-parameters follow similar selection strategy, namely balancing the model

performance and training efficiency. Our current selection for real-world datasets can be found in Tab. 19.

| Resolution | Syn Data-1 | | | | | Syn Data-2 | | | | |
|---|---|---|---|---|---|---|---|---|---|---|
| | 0.25 | 0.50 ⋆ | 0.75 | 1.00 | 1.25 | 0.25 | 0.50 ⋆ | 0.75 | 1.00 | 1.25 |
| ELBO ↑ | **18.45** | 18.33 | 16.62 | 15.39 | 15.22 | **22.08** | 21.42 | 19.46 | 16.80 | 16.25 |
| ER% ↓ | **40.67** | 41.72 | 42.74 | 45.60 | 45.93 | **46.52** | 46.85 | 48.78 | 49.60 | 51.25 |
| MAE ↓ | **2.28** | 2.32 | 2.36 | 2.53 | 2.62 | **2.45** | 2.52 | 2.67 | 2.88 | 2.85 |
| Time Cost (h) ↓ | 6.57 | 4.79 | 4.41 | 3.85 | **3.69** | 6.82 | 4.98 | 4.36 | 4.22 | **3.94** |

*Table 20.* Compare various time grid resolutions and their impact on the model performance for synthetic dataset. "⋆" indicates our current hyper-parameter selection for specific dataset. Other hyper-parameters and training sample size remain consistent.

| Max. Back-tracking Rounds | Syn Data-1 | | | | | | Syn Data-2 | | | | | |
|---|---|---|---|---|---|---|---|---|---|---|---|---|
| | 0 | 1 | 2 | 3 ⋆ | 4 | 5 | 0 | 1 | 2 | 3 ⋆ | 4 | 5 |
| ER% ↓ | 45.64 | 45.33 | 43.82 | 41.72 | 42.07 | **41.28** | 48.37 | 47.49 | 46.92 | 46.85 | 46.70 | **45.86** |
| MAE ↓ | 3.12 | 3.06 | 2.75 | 2.32 | 2.19 | **2.11** | 2.86 | 2.77 | 2.74 | 2.52 | 2.46 | **2.42** |

*Table 21.* Compare various maximum number of backtracking rounds and their impact on the model performance for synthetic dataset. "⋆" indicates our current hyper-parameter selection for specific dataset. Other hyper-parameters and training sample size remain consistent.

### E.2. Computing Infrastructure

All synthetic data experiments and real-world data experiments, including the comparison experiments with baselines, are performed on Ubuntu 20.04.3 LTS system with Intel(R) Xeon(R) Gold 6248R CPU @ 3.00GHz, 227 Gigabyte memory.

## F. Limitation and Broader Impacts

Our work has vast potential applications in the field of human-AI collaboration. The proposed approach enables timely and accurate inference of human mental events, as well as precise prediction of future human behavior. This will assist AI in providing timely, accurate, and useful assistance. For instance, it can aid elderly individuals with limited mobility in managing daily activities or help self-driving vehicles navigate roads more safely and smoothly.

One limitation of our current approach is the reliance on hand-crafted logic rule templates in the decoder. While these rules provide interpretability, they may introduce additional biases. To mitigate this limitation, we have already proposed a rule learning module via column generation, which can learn rule from an empty rule set. This approach enhances the model's flexibility by adapting to the nuances of the data and reduces the risk associated with manually introduced biases. We can explore more rule mining algorithms to improve the efficiency in the future work.

Additionally, the discretization of the timeline might introduce some noise when sampling the latent mental events. In real-world scenarios, establishing a well-defined and fine-grained discrete time grid can necessitate conducting numerous experiments. It's worth noting that choosing the interval for discretization is a tunable hyperparameter. We can explore methods to automate hyperparameter tuning to streamline this process and ensure optimal performance without the need for extensive manual experimentation.

