# OpenReview forum: "Evolving Minds: Logic-Informed Inference from Temporal Action Patterns"
_ICML.cc/2025/Conference — ICML 2025 poster_

### Official Review · Reviewer_BKJo · 2025-03-08

**Overall Recommendation:** 4

**Summary:**

The paper introduces a single framework to infer human intentions, predict future actions, and interpretable logical rules. The motivation is that human actions occur irregularly and are driven by unobserved mental states/intentions. To address this, the paper proposes a framework combining the temporal point process (TPP) and amortized variation Expectation-Maximization (EM) to model the bidirectional dynamics between irregular human actions and latent mental states. Logical rules are used as priors to guide TPP to build the relationship between intention and actions, reduce dependency on large datasets, and, ensure interpretability. The framework jointly optimizes the model parameters, logical rules, and inference networks. Experiments are conducted on synthetic and real-world datasets on action prediction task (next event prediction from history) and evaluated using error rate % and mean absolute error metrics and the framework shows a good improvement on both metrics for all datasets.

**Claims And Evidence:**

1. The claim that mental states drive human behavior and building a relationship between then can be helpful in predicting next actions has strong evidence as the framework performs well across all synthetic and real-world datasets.
1. The method optimizes the logical rules (initialized from scratch) and shows results that the rule learning module is able to learn the 4 ground truth rules in Figure 3.
2. Mental events are sampled based on the hazards and probabilistic sampling is able to sample reasonably within an error range as shown in Figure 3.
4. The paper also mentions the scalability of the approach and shows supported results in the supplementary.

**Essential References Not Discussed:**

No

**Experimental Designs Or Analyses:**

The evaluation criteria using error rate % and mean absolute error as metrics is valid and is being by the prior TPP method [1] as well. The paper also shows the accuracy of learning logical rules and the sampling efficiency of mental states which evaluates the method well. The comparison metric shows significant improvement when compared with baselines and further analysis on logical rules and sampling efficiency shows accuracy as well. In supplementary, the scalability of the approach is measured which supports the claim of the paper.

[1]. Zuo, S., Jiang, H., Li, Z., Zhao, T., and Zha, H. Transformer hawkes process. In International conference on machine learning, pp. 11692–11702. PMLR, 2020

**Methods And Evaluation Criteria:**

The objective of the paper is to improve next action prediction task by strengthening the relation between mental state and human actions.
The proposed method makes sense to address this objective. The evaluation criteria using error rate % and mean absolute error as metrics is valid and is being by the prior TPP method [1] as well. The paper also shows the accuracy on learning logical rules and the sampling efficiency of mental states which evaluates the method well.

[1]. Zuo, S., Jiang, H., Li, Z., Zhao, T., and Zha, H. Transformer hawkes process. In International conference on machine learning, pp. 11692–11702. PMLR, 2020

**Other Comments Or Suggestions:**

In Figure 3, in the fitted hazard graph, there is a typo in the label : Ture -> True.

**Other Strengths And Weaknesses:**

Strengths:
1. The motivation and contribution of the paper are strong and establish a relation between understanding latent human mental state and human actions.
2. The paper is written well and has a good explanation for the needed areas such as the subsection on comparison with VAEs.
3. The idea of logical rules is also helpful in reducing reliance on datasets and improving interpretability.

Weakness:
1. In Line 95, the authors claim that the rule discovery reveals novel and overlooked intention-action patterns. Is there any result/evidence in the paper that can support this?
2. As the idea of inferring human mental states seems to be a key contribution, the datasets used for experiments have very limited number of mental states such as EK-100 considers 2 states, hand-me-that considers 4, and car-following has a single mental state of human vehicle following human vehicle. Can the authors discuss why limited/few mental states in datasets are considered for experiments?

**Questions For Authors:**

1. Will the work be open-sourced?
2. What is the training time and how much resources does the framework require?

**Relation To Broader Scientific Literature:**

The key contribution of the paper of jointly inferring the human mental state and the human actions is helpful in the scientific literature of video understanding where it is essential to understand the context of the video and human intentions to anticipate future actions. The key contribution would especially be helpful for tasks such as action anticipation.

**Theoretical Claims:**

Yes

---

> ### Author Rebuttal · Authors · 2025-04-01
>
> We sincerely thank Reviewer BKJo for the insightful analysis and recognition of our work! We hope our responses listed below can address your concerns.
>
> **$\star$ Examples Revealing Newly Uncovered Rules**: We have reported a subset of temporal logic rules that have been identified as having real-world significance based on real-world datasets, and due to the page limitation, we put the results in the **Appendix D.4, Table 17**. These rules are learned by our model which are not provided as prior knowledge and easily overlooked. For the reviewer’s convenience, below we analyze one of the learned rules as a concrete example, explaining both its formal interpretation and real-world significance:
>
> $$PickUp \leftarrow WantToPickUp\ \wedge\ MoveTo\ \wedge\  Open, (WantToPickU\ before\ MoveTo), (MoveTo\ before\ Open), (Open\ before\ PickUp)$$
>
> This rule is learned from the Hand-Me-That dataset, capturing a sequential intention-action pattern: the user first develops an intention to pick up an object ($WantToPickUp$), then moves toward a destination ($MoveTo$), and opens the container ($Open$), ultimately leading to the target action ($PickUp$).
>
> The practical value lies in using this learned rule as guidance for mental state inference. When our model detects the user's WantToPickUp intention, the AI agent can proactively assist -- for instance, by opening the container in anticipation of the user's need. This inference and prediction capability can enhance human-agent interaction efficiency.
>
> **$\star$ Discussion on Limited/Few Mental States**: In real-world settings, mental states are typically **high-level and inherently sparse within specific contexts**. Our model operates under a **closed-world assumption**, focusing on constrained datasets tailored for specific tasks. As a result, the mental states in our experiments are confined to a **predefined domain-relevant set**—justifying our consideration of only a limited number of mental states for real-world applications. Encouragingly, our model **has the potential to adapt to scenarios with unknown or numerous mental states**. Inspired by [1], which uses vision-language models for predicate invention, we can also leverage LLMs to generate diverse mental states, circumventing the limitations of pre-defined states while producing arbitrarily numerous variations, before proceeding to rule learning and inference.
>
> [1] Liang, Y., Kumar, N., Tang, H., Weller, A., Tenenbaum, J. B., Silver, T., ... & Ellis, K. Visualpredicator: Learning abstract world models with neuro-symbolic predicates for robot planning. ICLR 2025.
>
> **$\star$ Training Time & Computing Infrastructure**: We have reported the training time for both synthetic datasets and real-world datasets as well as the computing infrastructure in our previous submission, and due to the page limitation, we put the results in the Appendix. Below we provide the section indexes for reviewer's convenience. The training time for varying sample sizes and the number of ground truth rules have been shown in **Appendix D.1**, for both synthetic datasets (**Figure 5**) and real-world datasets (**Figure 6**). Additionally, we have compared the impact of different hyper-parameter choices on training time and model performance in **Appendix E.1**. Details of the computational resources and computing infrastructure used in our experiments were provided in **Appendix E.2**.
>
> **$\star$ Open-Source**: Yes, we will release the codes for the final version.
>
> **$\star$ Correction for Typos**: Thanks for pointing it out! We have updated the legend in Figure 3 and made corresponding revisions to the final manuscript.

---

> > ### Comment · Reviewer_BKJo · 2025-04-06
> >
> > Thank you, authors, for providing the rebuttal and addressing my concerns! My concerns about the rule discovery revealing novel, overlooked rules, training time, open source, and typos were addressed well. The appendix has the results for new rule discovery and looks convincing. Regarding the discussion on limited and few mental states, having more fine-grained mental states would have added more strength to the work, but I believe the assumption of closed-world and sparse mental states works fine for the scope of the work.

---

### Official Review · Reviewer_w7nu · 2025-03-11

**Overall Recommendation:** 3

**Summary:**

This paper proposes combined logic-informed temporal point processes with amortized variational EM, allowing their method to infer underlying mental states reliably, even in low-data regimes.

**Claims And Evidence:**

Their experimental results show the effectiveness of this framework on some synthetic as well as real-world datasets.

However, I do have some concerns regarding scalability. First of all, I think it is an impressive result that the method works well in low-data regimes and is efficient through the use of EM and injection of priors as logic rules. However, I am less convinced by the current dataset sizes that there is evidence of scalability, despite the authors' pointing this out.

**Essential References Not Discussed:**

I am not aware of missing literature at this point.

**Experimental Designs Or Analyses:**

Overall the authors conduct experiments both on synthetic and real-world datasets to showcase their method.

**Methods And Evaluation Criteria:**

Yes, the baselines, datasets and the metrics used for evaluation all seem reasonable to me.

**Other Comments Or Suggestions:**

- Line 270 column 1: is computd --> is computed
- Line 272 column 2: nearconvergence --> near convergence or near-convergence
- Lines 292-292 colum1: model is easily adopt to large-scale --> model can easily be adopted ...?
- Line 282 column 2: other baseline attempt to predict **the** next event from history
- Line 303 column 2: In paticular --> in particular
- Line 403 column 2: faces scalability issue**s**

**Other Strengths And Weaknesses:**

I think overall this is a nice work, with the logic-informed priors offering interpretability in human-centric AI and the method showcasing good results in low-data regimes.

I do have some worries though regarding scalability claims -- I do think it is impressive that the framework works well in low-data regimes, I am bringing up scalability mainly because the authors mention it throughout the paper (that the method can scale), but the largest synthetic dataset used has 5K trajectories, and a small number of underlying rules (up-to 6 ground truth rules App D.1). I do not think this is enough evidence to support the scalability claims, I would be curious to hear the authors' thoughts on this.

Another weakness I find is in the presentation of the method: I think the overall setup of the method can be better motivated with some examples in the beginning of the Preliminaries section. For example, the explicit actions and mental events differentiation first comes up in Section 3.

**Questions For Authors:**

- As mentioned before, I would be curious to hear the authors' thoughts on how well their method can scale or at which point they realistically would expect it to not scale anymore.

- How does the method perform in unfamiliar/novel situations?

**Relation To Broader Scientific Literature:**

Learning to infer human intent, and predicting next actions from existing datasets is a relevant problem for human-centric AI and the authors propose a method that achieves good results in low-data regimes, with the logic rules offering additonal interpretability as well.

**Theoretical Claims:**

There aren't theoretical claims. The derivations used when presenting the method looked good to me.

---

> ### Author Rebuttal · Authors · 2025-04-01
>
> We thank Reviewer w7nu for the detailed analysis and insightful comments, which benefit us to further improve our paper! To address your concerns, we have prepared a detailed point-by-point response below.
>
> **$\star$ Scalability**: Your comments are very valuable! Yes. We acknowledge that our original statement was imprecise. A more accurate statement would be: "Our model demonstrates potential for scaling to large datasets."
>
> Compared with related methods, regarding our rule learning module, existing related approaches like TELLER [1] and CLNN [2] require only 600-2,400 training synthetic sequences. While event prediction models with latent states like AVAE [3] use up to 2,000 sequences. In comparison, our reported experiments utilize relatively large sample sizes.
>
> Moreover, our fine-grained model's computational complexity stems from three key factors:
>
> (1) The inherently combinatorial nature of rule learning.
>
> (2) Temporal discretization requiring inference at each tiny time grid for accurate continuous-time mental event inference.
>
> (3) Event density per sequence. In our scalability experiments, the most complex synthetic dataset comprises 5,000 sequences, with an average of 47.60 action events per sequence—totaling 238,000 events. This represents a relatively large-scale dataset.
>
> Therefore, scalability assessment must consider **not just sample size**, **but also rule learning complexity, temporal resolution, and event density** -- making our 5,000-sample experiments relatively large-scale under these demanding conditions.
>
> Considering the above key factors influencing scalability, we have added new experiments using Syn Data-2 and larger sample size to further assess the potential of our model. Please find **Table 1** in
>
> https://anonymous.4open.science/r/paper9060-F13F
>
> These new experimental results show that the model performance exhibits asymptotic stabilization with increasing dataset size, and our model still has potential to perform well in 10K+ sequences scale dataset within satisfactory training time cost.
>
> [1] Li, S., Feng, M., Wang, L., Essofi, A., Cao, Y., Yan, J., & Song, L. Explaining point processes by learning interpretable temporal logic rules. ICLR 2021.
>
> [2] Yan, R., Wen, Y., Bhattacharjya, D., Luss, R., Ma, T., Fokoue, A., and Julius, A. A. Weighted clock logic point process. ICLR 2023.
>
> [3] Mehrasa, N., Jyothi, A. A., Durand, T., He, J., Sigal, L., and Mori, G. A variational auto-encoder model for stochastic point processes. CVPR 2019.
>
> **$\star$ Illustrative Examples**: Thanks for your advice! We will incorporate additional illustrative examples in subsequent revisions to enhance comprehension. Following your suggestion **regarding the distinction between action and mental events, we added example** as:
>
> “Consider a person who intends to start exercising and later maintains regular strength training at the gym. This behavioral sequence involves: (1) a mental event (the unobservable intention to exercise) and (2) action events (the observable gym workouts). Crucially, mental events are internal and unobservable, while actions are external and directly observable—constituting what we actually perceive.”
>
> **$\star$ Unfamiliar Situation**:  Our rule-generation algorithm is inherently capable of learning logic rules from scratch, even in scenarios with limited prior knowledge, making it highly adaptable to unfamiliar situations. In practice, when encountering such scenarios, our system operates in two modes: If pre-trained on similar datasets, it directly transfers the well-trained model and takes the learned rules as priors to the new data; Otherwise, it autonomously learns new rules without requiring predefined logic rules, with the built-in backtracking mechanism further improving accuracy.
>
> We have assessed our method under two distinct unfamiliar conditions: **transfer adaptation (for domain-related cases)** and **fully unfamiliar application (for entirely unseen datasets)**, with comparative results presented in **Table 2** in
>
> https://anonymous.4open.science/r/paper9060-F13F
>
> These new experimental results confirm our model **maintains robust performance in fully unseen scenarios**. When **pre-trained** on similar datasets, the learned rules **boost both training efficiency and final model performance**.
>
> **$\star$ Correction for Typos**: Thanks for your careful review! All typos have been corrected and will be updated into the final version.

---

> > ### Comment · Reviewer_w7nu · 2025-04-06
> >
> > Thank you for the new experiments and for the clarifications. I will keep my current score.

---

### Official Review · Reviewer_JJ34 · 2025-03-11

**Overall Recommendation:** 3

**Summary:**

The paper presents an amortized variational EM framework for understanding human mental states by modeling the relationship between actions and hidden mental events over time. Some innovations include using logic rules as priors to improve interpretability and approximating the posterior distribution of latent mental states by discrete-time renewal process.

**Claims And Evidence:**

The paper makes several claims about the effectiveness of their proposed method, and overall, these are supported by experimental evidence.

**Essential References Not Discussed:**

NAN

**Experimental Designs Or Analyses:**

The experiments on multiple datasets demonstrate improved event prediction performance, and the baselines from three categories—Neural TPP, Rule-Based, and Generative models—provide fair comparisons. Additionally, the ablation study on backtracking supports its effectiveness.

**Methods And Evaluation Criteria:**

The proposed methods and evaluation criteria generally align well with the problem of inferring human mental states and predicting future actions from temporal data. The use of two strategies tailored to different scenarios is useful. One question regarding autonomous rule learning for data-rich domains: in Table 17, the elements of temporal logic rules, such as PickUp and WantToPickUp, are still predefined, right?

**Other Comments Or Suggestions:**

See above.

**Other Strengths And Weaknesses:**

See above.

**Questions For Authors:**

See above.

**Relation To Broader Scientific Literature:**

This work builds on existing work by combining logic reasoning with neural networks, dynamically discovering rules while balancing model flexibility and explainability.

**Theoretical Claims:**

No formal theoretical claims are provided in this work.

---

> ### Author Rebuttal · Authors · 2025-04-01
>
> Thank you for your positive feedback!
>
> Regarding your question:
>
> _"One question regarding autonomous rule learning for data-rich domains: in Table 17, the elements of temporal logic rules, such as PickUp and WantToPickUp, are still predefined, right?"_
>
> Yes, in our current framework, predicates (i.e., _"elements of temporal logic rules"_ as shown in Appendix D.4, Table 17) are drawn from a predefined candidate pool. This reflects a **closed-world assumption**, where all possible actions and mental states are specified in advance.
>
> However, our approach can be extended to an **open-world setting** by incorporating **predicate invention mechanisms**. Inspired by prior work [1] on using vision-language models for predicate generation, we can leverage LLMs to handle novel situations—first prompting them to generate diverse candidate actions and mental states, and then integrating these into the rule-learning and inference process.
>
> [1] Liang, Y., Kumar, N., Tang, H., Weller, A., Tenenbaum, J. B., Silver, T., ... & Ellis, K. Visualpredicator: Learning abstract world models with neuro-symbolic predicates for robot planning. ICLR 2025.

---

> > ### Comment · Reviewer_JJ34 · 2025-04-03
> >
> > Thank you for your clarification. I have no further questions and will keep my score.

---

### Decision · Program_Chairs · 2025-05-01

**Decision:**

Accept (poster)

**Comment:**

All reviewers voted accept based on (1) the paper proposes a reasonable idea, (2) the experiments are solid, (3) the work helps reduce reliance on datasets and improve interpretability, etc., which I mostly agree with. Some concerns were raised but have been properly addressed.